

# High-resolution predictions of ground ice content for the Northern Hemisphere permafrost region

Olli Karjalainen[1], Juha Aalto[2,3], Mikhail Z. Kanevskiy[4], Miska Luoto[2], and Jan Hjort[1]

[1]Geography Research Unit, University of Oulu, 90014, Oulu, Finland

[2]Department of Geosciences and Geography, University of Helsinki, 00014, Helsinki, Finland

[3]Finnish Meteorological Institute, 00101, Helsinki, Finland

[4]Institute of Northern Engineering, University of Alaska Fairbanks, Fairbanks 99775-5910, Alaska, U.S.A.

*Correspondence to*: Olli Karjalainen (olli.karjalainen@oulu.fi)

**Abstract.** Ground ice content of the Arctic soils largely dictates the effects of climate change-induced permafrost degradation and top ground destabilization. The current circumarctic information on ground ice content is overly coarse for many key applications, including assessments of hazards to Arctic infrastructure, while detailed data are restricted to very few regions. This study aims to address these gaps by presenting spatially comprehensive data on pore and segregated ground ice content across the Northern Hemisphere permafrost region at a 1-km resolution. First, ground ice content datasets (n=437 and 380 1-
km grid cells for volumetric and gravimetric ice content, respectively) were compiled from field observations over the permafrost region. Spatial estimates of ground ice content in the near-surface permafrost north of the 30th parallel north were then produced by relating observed ground ice content to physically relevant environmental data layers of climate, soil, topography, and vegetation properties using a statistical modelling framework. The produced data show that ground ice content varies substantially across the permafrost region. The highest ice contents are found on peat-dominated Arctic lowlands and
along major river basins. Low ice contents are associated with mountainous areas and many sporadic and isolated permafrost regions. The modelling yields relatively small prediction errors (a mean absolute error of 13.6 % volumetric ice content) over evaluation data and broadly congruent spatial distributions with earlier regional-scale studies. The presented data allow the consideration of ground ice content in various geomorphological, ecological, and environmental impact assessment applications at a scale that is more relevant than previous products. The produced ground ice data are available in the
supplement for this study and at Zenodo https://doi.org/10.5281/zenodo.7009875 (Karjalainen et al., 2022).

## 1 Introduction

The distribution and abundance of ground ice in the Northern Hemisphere permafrost soils remains one of the least known
components of the cryosphere (Jorgenson and Grosse, 2016; Schuur and Mack, 2018). Notwithstanding, ground ice content is a key factor in assessing how permafrost regions may evolve in changing environmental conditions (Gilbert et al., 2016; Nitzbon et al., 2020; Smith et al., 2022). Ground ice content is central for the thermal response of permafrost to climate change because it affects the rate at which frozen ground thaws due to the latent heat required to melt ice (Lee et al., 2014;





Couture and Pollard 2017; Cai et al., 2020). The melting of ground ice, in turn, can lead to changing soil volume and inflict surface subsidence and hydro-ecological alterations (Osterkamp et al., 2009; Zhao et al., 2020).

A general term "ground ice" refers to all types of ice contained in freezing and frozen ground; ground ice occurs in pores, cavities, voids or other openings in soil or rock (van Everdingen, 1998). Three major types of ground ice exist: pore, segregated, and massive ice. Pore ice (sometimes termed interstitial or "cement" ice) fills pores of soils and rocks; it forms where pore water freezes in place and is the most common type of ground ice (van Everdingen, 1998; Murton, 2013; French,

2018). Segregated ice is formed by the migration of pore water towards the freezing plane where it forms lenses or layers of ice. Thin lenses and layers of segregated ice form cryostructures of frozen soils (Fig. 1) while thick layers (tens of cm and more) should be described as massive-ice bodies. Segregated ice forms in a variety of materials but water-saturated fine-grained soils are most favorable for its development (van Everdingen, 1998; Murton, 2013; French, 2018). Massive ice is a comprehensive term used to describe large masses of ground ice, including ice wedges (the most common type of massive

ground ice), pingo ice, buried ice, and thick layers of segregated ice (van Everdingen, 1998).

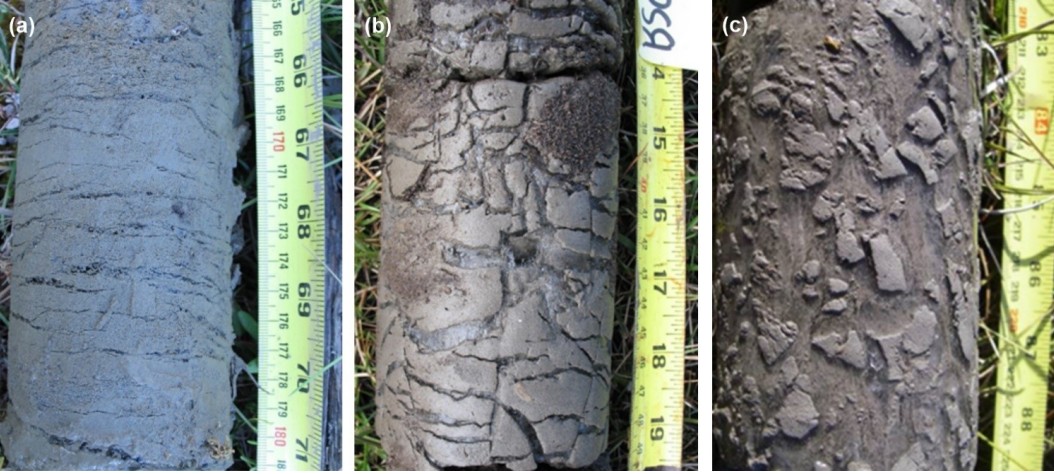

**Figure 1: Typical cryostructures of frozen soils from samples of frozen cores collected in Northern Alaska: Layered cryostructure in silty sand with some coarse inclusions, ice lenses <0.2 cm thick; borehole D170-2, Coldfoot area**

**along the Dalton Highway; gravimetric ice content (GIC) of 33.5 % at the depth of 173–180 cm (a); Reticulate cryostructure in silt with peat inclusions, ice lenses <0.5 cm thick; borehole BSC-20, Beaufort Sea coast near Cape Halkett; 80.6 %GIC, volumetric ice content (VIC) of 64.9 % at the depth of 38–48 cm (b); Ataxitic (suspended) cryostructure in silty clay, soil aggregates <1.5 cm across are suspended in ice; borehole CAHA-E1, Beaufort Sea coast near Point McLeod; 106.6 %GIC at the depth of 211–223 cm (c).**


Accumulation of ground ice is a result of climate-driven thermal, hydrological and geomorphic processes in the soils as well as geologic and glacial history (Dredge et al., 1999; French and Shur, 2010; Murton, 2013; Walvoord and Kuryluk, 2016; French, 2018). Ground ice content strongly depends on mechanisms of permafrost formation. Of the two main types, epigenetic permafrost forms through lowering of the permafrost base in previously deposited sediment or other earth

material, while syngenetic permafrost forms through rise of the permafrost table during the deposition of additional sediment or other earth material on the ground surface (French and Shur, 2010; Shur et al., 2011; Murton, 2013; French, 2018).



Coarse-grained material favors the evaporation and sublimation of moisture to the atmosphere (Bockheim and Tarnocai, 1998), whereas finer sediments have a higher moisture retention capacity and frost susceptibility (Kokelj and Burn, 2005; French, 2018) and as such potentially higher ground ice contents. Notwithstanding, Lacelle et al. (2022) point out that in

many Arctic locations ground ice contents of coarser sediments are not necessarily lower than those in frost-susceptible soils. Peatlands and organic-rich soils often accommodate very high ice contents due to the high porosity and low density of peat (Smith et al., 2012; Fan et al., 2021; Lacelle et al., 2022).

Schuur and Mack (2018) argued that ground ice alongside permafrost temperature and organic carbon content dictate the consequences of changing permafrost for ecosystems and society. The current and future thermal state of permafrost

(Chadburn et al., 2017; Aalto et al., 2018; McGuire et al., 2018; Obu et al., 2019; Burke et al., 2020; Li et al., 2022; Smith et al., 2022) and its organic carbon content (Hugelius et al., 2020, Mishra et al., 2020, 2021) have been studied from regional to global scales but the information on ground ice distribution have remained poorly resolved. The IPCC Special Report (Meredith et al., 2019) concludes that the Northern Hemisphere ground ice volume and its distribution are only known with medium confidence and with no recent updates at the circumpolar scale. Circumarctic assessments of permafrost-related

hazards to infrastructure, for example, have been limited by the inaccurate or coarse-scaled depictions of local ground ice content (Hjort et al., 2018, 2022; Streletskiy et al., 2019). Moreover, Lacelle et al. (2022) argued that substantial areas in the current ground ice maps have underestimated ground ice abundances owing to the often-presumed association of low ice contents and coarse-grained sediments which may not be universally applicable.

The Circum-Arctic Map of Ground ice conditions (Brown et al., 1997), known as the International Permafrost Association

(IPA) Permafrost Map, is a global observation-based map that continues to facilitate large-scale permafrost studies. The IPA map distinguishes only a few classes of estimated visible ice content, and its low spatial and thematic resolution is a potential bottleneck for reliable Arctic impact assessments. Recent broad-scale ground ice mapping efforts remain rare. O'Neill et al. (2019) executed a paleogeographic mapping of ground ice abundance for the entire Canada by using geospatial data on surficial materials, glacial history, palaeovegetation, and modern permafrost distribution. Other approaches use permafrost

landforms as proxies of ice-rich environments (Jorgenson et al., 2014; Walvoord and Kuryluk, 2016; Schuur and Mack, 2018; Karjalainen et al., 2020). Ground ice content has also been simulated by process-based land surface models. For example, Saito et al. (2021) used a land surface model to simulate long-term ground ice budget across the circumarctic permafrost region. Others have used the IPA map (Brown et al., 1997) to derive excess ice estimates at coarse resolution for land surface model simulations (e.g., Lee et al., 2014; Cai et al., 2020). A disadvantage of process-based models is that their

spatial resolution (~10–100 km) limits depicting the fine-scale spatial variation in ground ice content. To consider sub-grid variability, some recent applications use a tiling approach in which grid cells in land surface models are partitioned by environmental similarity (Nitzbon et al., 2020, 2021).

Here, we present datasets of harmonized ground ice content observations and statistical model predictions of the distribution of ground ice content for pore and segregated ice at a ~1 km resolution across the Northern Hemisphere permafrost region.

The observational datasets (volumetric ice content (VIC) n=437 and gravimetric ice content (GIC) n=380) serve as previously unavailable validation data for regional to permafrost region-wide applications. The predictions enable estimating the distribution of absolute ground ice values instead of often used nominal outputs (e.g., low, moderate, and high ice content) at high spatial resolution. In addition, we compare our predictions to previous ground ice assessments. Our spatial ground ice data offer new information on environmental conditions in the Arctic and present an important step towards

developing improved future projections of changing permafrost landscapes.



## 2 Methods and input data

### 2.1 Ground ice data compilation

Data on ground ice content are frequently recorded during permafrost studies and geotechnical investigations in the permafrost region (e.g., Johnston, 1981). Usual method is to drill a borehole to extract soil cores from which ice content can
be either estimated visually or measured using protocols, which are based on either the volume or mass of the ice contained in the sample. VIC is the ratio of the volume of ice in the sample to the volume of the sample and is expressed as a percentage (van Everdingen, 1998). It is scaled between 0 and 100 % and gives a clear measure of how much ice is contained in the ground (hereafter, the unit %VIC is used). Quantification of VIC requires high-quality samples and is a relatively time-consuming, and thus GIC is more often determined in the field. GIC can be computed by weighing the frozen
(or thawed) and dry (e.g., oven-dried) masses of the sample. Dry-weight GIC is the ratio of the mass of ice in the sample to its dry mass (van Everdingen, 1998). Ice mass corresponds to the mass of water lost in the drying. Rarely used wet-weight GIC, in turn, is the ratio of ice mass in the sample to its wet mass (Phillips et al., 2015). The dry and wet GIC are expressed as a percentage (%GIC), and while dry GIC can greatly exceed 100 %, wet GIC is scaled between 0 and 100 %.

The data on ground ice contents across the Northern Hemisphere permafrost region were compiled from published databases
and articles (Appendix E, Supplementary Data 1). From these data, we used measured ice content values at all applicable core sections (i.e., samples) to compute average ice content for near-surface permafrost. In addition, we extracted ground ice values from graphs and plots whenever it was possible to determine the reported ice content with an accuracy of ca. 1 %VIC/GIC and computed average values for the covered depth interval as previously. Values reported for entire cores were also acquired from text provided that the values were documented with similar accuracy and for identifiable boreholes with
verifiable locations and appropriate depths of sampling. Each core was located with at least a few hundred meters accuracy. Measurements from the 1980s to present day were included to maximize the geographical coverage of modelling data. Whatsoever, regardless of the time of observation, the same strict inclusion criteria were applied for all data. In total, VIC dataset was based on 845 individual cores while 670 cores were utilized in the GIC dataset (1–34 cores per 30 arc-second grid cell).

To achieve maximal comparability between ground ice observations across the Northern Hemisphere, average ice content for each core was computed for the first five meters below the top of permafrost (see O'Neill et al., 2019). This depth was considered to contain most of the ice (French, 2018) as documented in numerous locations across the permafrost domain (e.g., Brown, 1968; Lewellen, 1973; Pollard and French, 1980; Seguin and Frydecki, 1994; Yin et al., 2017; Paul et al., 2020) and to have the most relevance concerning the Earth surface processes. In less than ten cases where a single average
ice content for an entire core was provided in the source data, the five-meter constraint could not be applied, and values representing longer profiles were included. Whereas such deep profiles can be considered to yield a reliable estimate of the average ice content at the site, very short samples are more prone to have spurious values owing to the vertical variation in cryostratigraphic soil units. Therefore, to ensure the representativeness and comparability of the observations we excluded all sites that were represented with a single sample of less than 20 cm in length. This threshold was chosen to not omit too
many observations and consequently lose geographical coverage of the modelling data (20 grid cells were omitted).

To account for the vertical variation of ice contents encountered within one borehole, we computed a weighted average of ice contents based on the length of the core sections (Subedi et al., 2020) or the number of measurements per soil unit pre-defined in the source data. Also, to avoid pseudoreplication, if more than one observation were located inside the same 30 arc second grid cell (in cases, such as borehole transects), a similar weighted average was computed for the grid cell based
on individual core lengths (Fig. 2). Measurements (either entire cores or individual samples) recorded from massive ice (e.g.,





ice wedges, pool ice, or buried ice) were excluded to avoid the inflation of the site's ice content due to the extremely high volumetric or gravimetric values associated with massive ice. In this study, we consider ground ice content mainly due to pore and segregated ice (see Harris and Murton, 2005). Moreover, sites that had reportedly undergone fires during the last few years were not included.

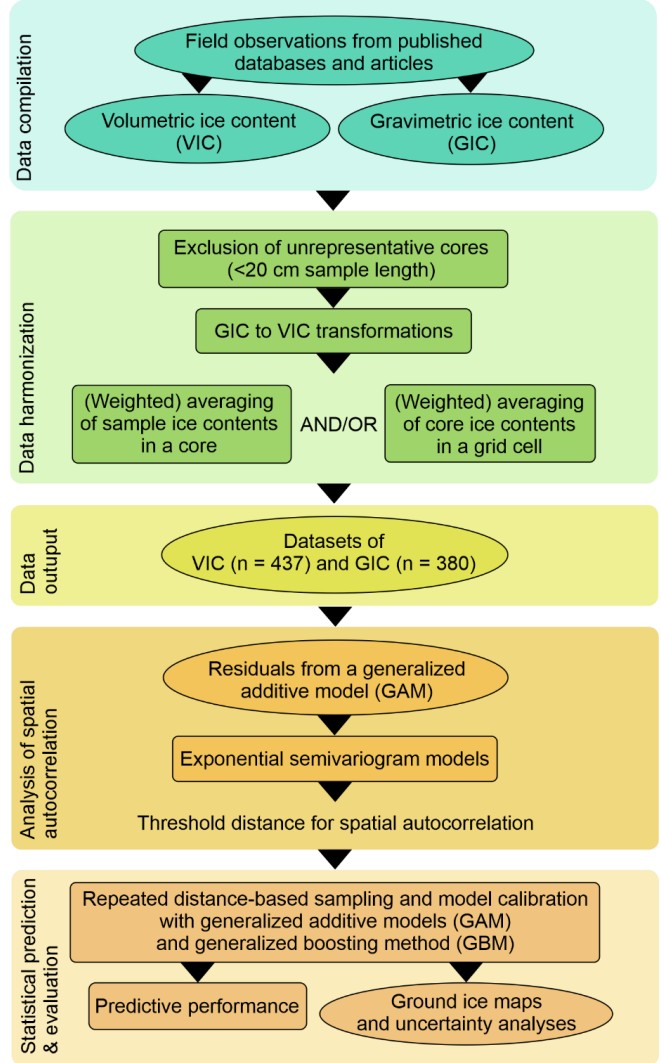


**Figure 2: Conceptual diagram of ground ice content dataset compilation and statistical modelling procedures that were performed to produce spatial predictions of ground ice content, i.e., ground ice maps.**

**2.2 Gravimetric to volumetric ice content transformations**

More GIC than VIC measurements were initially available in the data sources. However, as VIC has more relevance when estimating thaw settlement characteristics of the ground and permafrost sensitivity to disturbance and thawing, we transformed all GIC values to VIC (Shur et al., 2021, Supporting Information). Another benefit of VIC is that it is a readily comprehensible measure of ground iciness that is comparable across the permafrost domain while weight-based GIC values




have vastly different ranges in low- and high-density soil materials. The transformations also made it possible to include all the compiled ground ice observations across the permafrost domain. Considering permafrost thaw-related processes, the most important ice content measure would be excess ice content, i.e., the amount of ice that exceeds the pore volume of the soils (van Everdingen, 1998; Cai et al., 2020; Nitzbon et al., 2020). Excess ice data, however, are very limited at circumarctic scale and typically do not include necessary parameters for converting GIC or VIC to excess ice. Visible ice contents were similarly excluded because the measure does not include pore ice (see Paul et al., 2020).

**2.2.1 Bulk density transformation.** The dry bulk density (DBD) of solid soil materials, derived as the ratio between the dry mass of soil material and the total volume of the sample, was used to translate gravimetric ice content to volumetric whenever DBD was reported in the source data. The transformation was done for the GIC values associated with individual samples before computing the core average. The following equation was utilized (adapted from Bilskie, 2001; Jelinski et al., 2019) to estimate VIC based on GIC and dry bulk density Eq. (1):

$$VIC=GIC*DBD/G_i,$$

(1)

where GIC is the gravimetric ice content, DBD is the dry bulk density of soil material, and $G_i$ is the density of ice (0.9). We estimated the potential error produced in the transformation by comparing the measured and transformed values from 911 individual core sections from those source data that had reported GIC, VIC and DBD for each sample. The transformed VIC against the measured VIC had a mean absolute error of 2.1 %VIC (standard deviation 11.3 %VIC).

**2.2.2 Specific gravity transformation.** When information on ice volume or soil DBD was not provided in the source data, GIC was transformed to VIC using standard empirical specific gravity values. The values were manually defined for each observation site based on either soil descriptions, grain size analyzes, profile diagrams and photographs, or (if any site-specific information could not be found) regional descriptions of soils that were provided in the data source. We differentiated those cores with 1) mostly mineral soil material (>90 % of the length of the vertical profile), 2) a mix of organic and mineral materials (i.e., less pronounced difference in associated material proportions), and 3) mostly organic soils (>90 %). Then, we attributed specific gravity values for the three classes using values of 2.5 for mineral soils, 2.0 for mixed, and 1.5 for organic soils (peat), and applied the following equation (Johnston, 1981):

$$VIC=(GIC*G_s/G_i)/(1+GIC* G_s/G_i)$$

(2)

where VIC is volumetric ice content, GIC is gravimetric ice content, $G_s$ is the specific gravity of soil material, and $G_i$ is the density of ice (0.9). When there was more than one sample from a core (or more than one core from a site) the transformation was done for each sample (or core) before computing the site average. Some of the available GIC observations were reported as wet GIC values and were translated to dry GIC using the following equation (Phillips et al., 2015) prior to transformations:

$$dry\ GIC=wet\ GIC/(1 – wet\ GIC)$$

(3)

The successfulness of the specific gravity transformation was evaluated by comparing the estimated VIC values from Eq. (2) to observed VICs for the sites for which both GIC and VIC were provided in the source data (n=182). On average, absolute difference was 6.2 %VIC (standard deviation 6.3 %VIC). In the final VIC modelling dataset, 224 grid cells values were computed from field observations initially provided as VIC, 13 grid cell values were derived from GIC with Eq. (1) and 200



with Eq. (2). Both with the bulk density- and specific gravity-based transformations, the error estimates are representative of the respective subsamples of the full dataset and may be different for the remaining observations for which the error statistics cannot be computed owing to the lack of either GIC, VIC or DBD.

## 2.3 Environmental predictors

We use geospatial data on soil, climate, topography, and vegetation characteristics to account for the variation in relevant environmental factors for ground ice content (Table 1). Gravimetric contents of clay (g kg⁻¹), silt (g kg⁻¹), sand (g kg⁻¹) and soil organic carbon (dg kg⁻¹), and volumetric coarse fragments (cm³ dm⁻³) contents in the soils were derived from 250 m spatial resolution SoilGrids 2.0 data (Poggio et al., 2021) and resampled to 30 arc second resolution using bilinear sampling. We used the values for the depth interval 100–200 cm assuming that in many cases the top meter of soils roughly coincidences with the active layer with no perennial ground ice. We also considered the potential for ground ice aggradation by including spatial data on probability of R horizon (bedrock) occurrence within the first 200 cm below surface (Shangguan et al., 2017).

Table 1. Environmental predictors considered in the modelling. All 14 predictors were used in a 30 arc-second resolution.

| Predictor (abbreviation) | Unit | Source |
|---|---|---|
| Fraction of clay (Clay) | g kg⁻¹ | |
| Fraction of silt (Silt) | g kg⁻¹ | |
| Fraction of sand (Sand) | g kg⁻¹ | Poggio et al., |
| Fraction of fine soil materials (sum of Clay and Silt) | g kg⁻¹ | 2021 |
| Coarse fragments (Coarse) | cm³ dm⁻³ | |
| Soil organic carbon content (SOC) | dg kg⁻¹ | |
| R horizon occurrence (Rhorizon) | probability of occurrence 0–100 | Shangguan et al., 2017 |
| Freezing-degree days (FDD) | °C-days in a year, average for 1950–2000 | |
| Thawing-degree days (TDD) | °C-days in a year, average for 1950–2000 | Hijmans et al., |
| Sum of solid precipitation (Snowfall) | mm in a year, average for 1950–2000 | 2005 |
| Sum of liquid precipitation (Rainfall) | mm in a year, average for 1950–2000 | |
| Coverage of open water bodies (WaterCover) | % | Defourny et al., 2016 |
| Topographic wetness index (TWI) | index | Böhner and Selige, 2006 |
| Normalized difference vegetation index (NDVI) | index | Didan et al., 2015 |



WorldClim data (Hijmans et al., 2005) were used to compute annual average freezing- (FDD, °C-days) and thawing-degree days (TDD), as well as to estimate annual average sums of snow (Snowfall, mm/yr) and rain (Rainfall, mm/yr) precipitation for the period 1950–2000. Snowfall represents the sum of precipitation during months with average temperature below 0 °C
and rainfall for those above 0 °C. The effects of water bodies, whether hydrothermal or due to the spatial association between thermokarst lakes and ice-rich environments, were accounted for by computing water coverage in a grid cell based on 150 m pixels in the global dataset by Defourny et al. (2016). Moreover, we used the GMTED2010 digital elevation model (Danielson and Gesch, 2011) to compute a topographic wetness index (TWI) in SAGA GIS (System for Automated Geoscientific Analyses, Conrad et al., 2015) with the SAGA Wetness Index tool (Böhner and Selige, 2006). TWI provides a
measure of soil moisture potential based on the flow and accumulation characteristics of drainage areas defined from the elevation values. Finally, we produced a normalized difference vegetation index (NDVI) from MODIS data (Didan, 2015). Pixel values show the average NDVI for the summer period (June–August) for the period 2000–2014.

Values from each environmental predictor were extracted for each grid cell assigned with VIC or GIC. To avoid omitting too many observations from the modelling datasets, we extracted values from the adjacent pixel in the cases where the borehole
location did not situate inside any pixel. Other than for the climate predictors, we did not extract values further than the adjacent pixel owing to the assumed high pixel-scale variation in soil properties, for example, but excluded the observation due to missing data. After the grid cell-wise averaging procedures, exclusions of cores based on a single <20 cm sample, and omitting grid cells without extractable environmental predictor pixel values, 437 (VIC) and 380 (GIC) unique ~1-km grid cells comprised the modelling datasets (Fig. 2, Supplementary Data 1).


### 2.4 Statistical modelling

Statistical models were used to link the VIC observations and corresponding environmental conditions at 30 arc-second grid cells and to predict ground ice content across the study area. A regression-based generalized additive model (GAM) was selected owing to its capability of both fitting smooth non-linear relationships and producing sufficiently general model
structures for predicting outside the modelling data domain (Guisan et al., 2002). In addition, generalized boosting method (GBM) was employed. GBM can fit highly non-linear relationships while automatically accounting for interaction effects between environmental predictors (Elith et al., 2008). GBM is more data-driven and reportedly capable of achieving low prediction errors with relatively small datasets in circumarctic contexts (e.g., Karjalainen et al., 2019, 2020; Mishra et al., 2020; Virkkala et al., 2021). Models were run in R (R Core Team 2021, R version 4.1.0) with the packages mgcv (Wood, 2011, version 1.8.31) for GAM and dismo (Hijmans et al., 2017, version 1.1.4) for GBM. Model fitting procedures are
detailed in the Appendix A. No exceedingly strong correlations were recorded among the selected predictors in the models (each <|0.7| Spearman's rhoo, Fig. A1).

The predictive performance of the models was evaluated by using a 100-fold distance-blocked cross validation (Roberts et al., 2016). In the procedure, the models calibrated at each 100 rounds were evaluated against spatially independent
evaluation datasets. The rationale behind this approach is that unaccounted spatial autocorrelation, i.e., the tendency of nearby observations to resemble each other more than those further, could yield too optimistic estimates of the model's predictive performance (Dormann et al., 2007). The distance at which spatial autocorrelation effect dissipated among the residuals from a GAM was first determined using exponential semivariogram model (Fig. B3). Then we used the distance estimate (45 km) with an R function *zerodist2* (sp package, Pebesma et al., 2005) to split the original data randomly to
calibration and evaluation datasets between which no spatial autocorrelation would exist. On average, 251 VIC observations were used in the 100 calibration runs (Fig. A2) and 44 in the evaluation.

To quantify the predictive performance, mean error (ME), mean absolute error (MAE), root mean square error (RMSE) and coefficient of determination ($R^2$) were calculated between predicted and observed ice content for both calibration and evaluation datasets. For external evaluation, the modelled spatial patterns of VIC were compared to earlier broad-scale

mapping efforts. However, rather than the general patterns of ground ice distribution, a key factor in the light of permafrost dynamics is the abundance of ice content at a location. Some more detailed local to regional estimates of ground ice abundance exist and were here semi-quantitatively compared to our VIC predictions.

## 3 Results

### 3.1 Northern Hemisphere distribution of ground ice content

The average observed VIC and GIC values for the Northern Hemisphere permafrost region are 61 % (range 0–96 %) and 176 % (0–1,717 %), respectively (Fig. 3). On average, the VIC and GIC values are representative of a 235 and 236 cm soil column (ranging from 20 to 2,500 cm) and have their average top and bottom at 89 and 324 cm, and 81 and 317 cm, respectively. The representative soil column lengths and ground ice contents have statistically significantly negative correlation; for VIC (n=423) Spearman's rho is –0.18 (p<0.001) and for GIC (n=367) –0.18 (p<0.001).


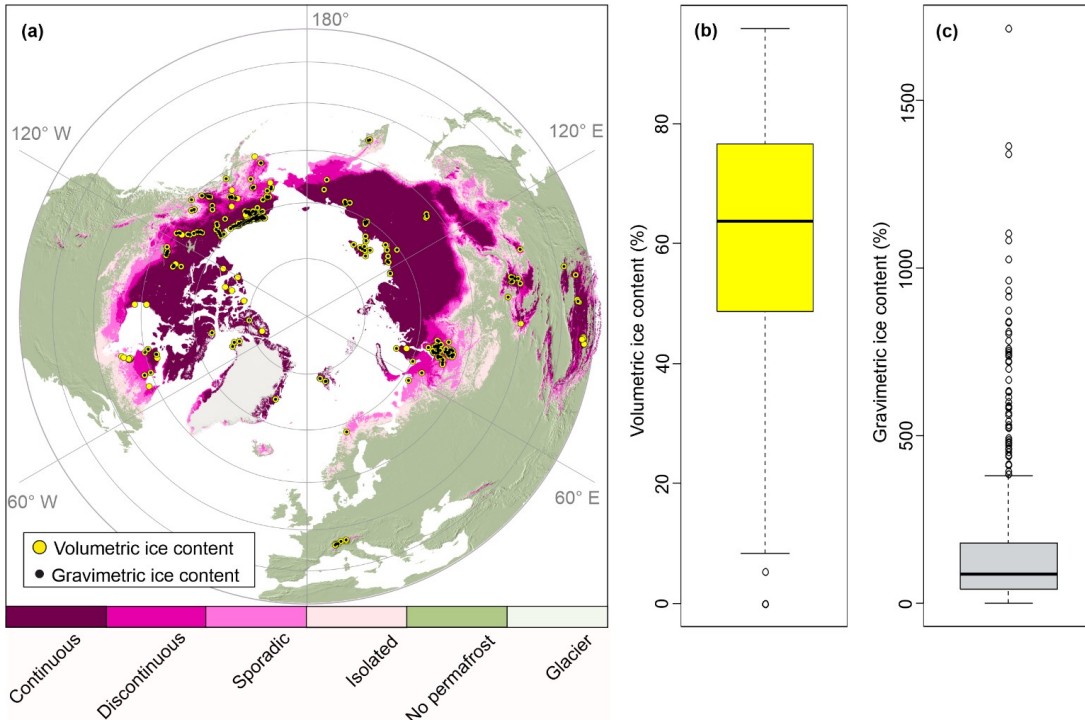

**Figure 3: Compiled ground ice observations across the Northern Hemisphere permafrost region (a) and boxplots summarizing the distributions of volumetric (b) and gravimetric ice content (c) values. Thick black line is the median and the top and bottom of the box coincidence with 75th and 25th percentiles, respectively. Circle symbols represent**

**outlier values which are more than 1.5 times the interquartile range away from the 25th or 75th percentiles. Permafrost zonation is from Obu et al. (2019).**

The predicted distribution of VIC shows strong spatial variation at different scales across the permafrost domain (Fig. 4a). The areas with the highest VIC correspond with the low-lying drainage basins of major Arctic rivers, such as Lena, Kolyma and Yenisei in Russia, and Yukon and Mackenzie in North America. Extensive ice-rich permafrost regions are also found

across the vast peatlands in West Siberia and south of the Hudson Bay. Also pronounced are certain coasts of the Arctic Ocean. Mountainous areas are the lowest in ice content. Relatively low ice contents are also predicted for many of the Canadian Arctic Islands and the western coast of Greenland while the eastern coast is more ice rich.

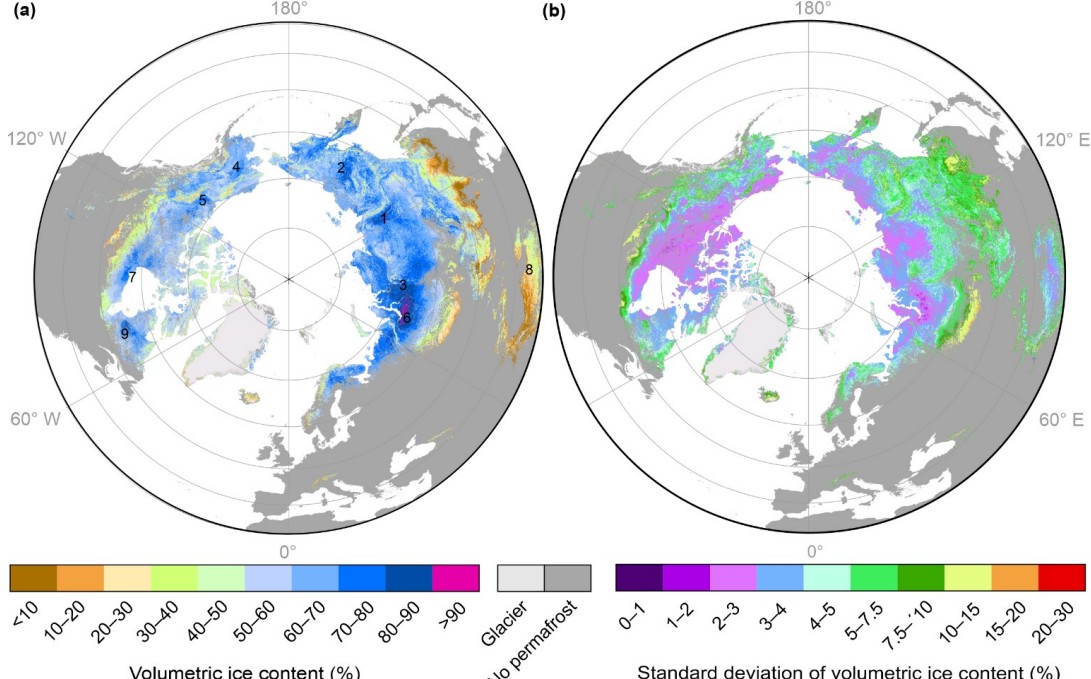

**Figure 4: Predictions of volumetric ice content (a, %VIC) and their standard deviation (b) for the Northern Hemisphere**
**permafrost region based on 100 generalized additive modelling (GAM) runs. Locations mentioned in text are numbered: 1) Lena River, 2) Kolyma River, 3) Yenisei River, 4) Yukon River, 5) Mackenzie River, 6) West Siberian Lowland, 7) Hudson Bay Lowland, 8) the Tibetan Plateau, and the central Ungava Peninsula (9). Permafrost extent is based on Obu et al. (2019). More detailed regional maps are presented in Figure D5. The results for generalized boosting method (GBM) are presented in Figures D6–7.**


The predicted average VIC for the permafrost region is 55.9 %. The Tibetan Plateau has clearly lower ice contents than the circumarctic area (on average 24.0 %VIC for the entire permafrost region of China) but also has multiple regions with VIC above 40–50 %. Among other key permafrost regions, Russia has the highest average VIC value (62.0 %) and is followed by the United States (57.7 %), Canada (53.9 %), Greenland (48.1 %) and Mongolia (26.8 %). Prediction variation is the lowest

in the high-Arctic regions. The standard deviation among 100 predictions is below 4 %VIC over most of the continuous permafrost regions with the notable exceptions Lena River valley and Yakutian lowlands (Fig. 4b). Highest variation occurs in mountainous regions (~5–10 %VIC) and those associated with isolated permafrost (~10–20 %VIC).



### 3.2 Model performance

The AIC-optimized VIC models include eight predictors: FDD, TDD, Snowfall, Rhorizon, fraction of fine soil materials,
NDVI, TWI and WaterCover. The predictive performances of the models ($R^2$) show a clear drop when calibrated models are
used to predict VIC at sites in the spatially independent evaluation datasets while prediction errors show a more modest
increase (Table 2). Standard deviation of errors and $R^2$ also increase in the evaluation setting. Overall, predicted VIC values
have a negligible positive bias of 0.3 %VIC (GAM). GAM and GBM yield similar statistics for evaluation datasets while
GBM has a better match with the calibration data. For example, GBM predicts better over the calibration datasets, but GAM
nevertheless has a similar $R^2$ for evaluation data (i.e., is more robust).

Table 2. Predictive performance of generalized additive modelling (GAM) and generalized boosting method (GBM) in volumetric ice
content (VIC) modelling based 100-fold distance-blocked cross validation. Presented are the average and [standard deviation] of mean
error (ME), mean absolute error (MAE), root mean squared error (RMSE) and coefficient of determination ($R^2$).

|  | ME (%VIC) | | MAE (%VIC) | | RMSE (%VIC) | | $R^2$ | |
|---|---|---|---|---|---|---|---|---|
|  | Calibration | Evaluation | Calibration | Evaluation | Calibration | Evaluation | Calibration | Evaluation |
| GAM | -0.02 [0.02] | 0.3 [3.2] | 12.0 [0.5] | 13.6 [1.7] | 15.3 [0.6] | 17.0 [2.0] | 0.53 [0.04] | 0.29 [0.17] |
| GBM | <0.01 [0.01] | 1.1 [3.1] | 7.4 [1.1] | 13.8 [1.6] | 9.7 [1.3] | 17.3 [2.0] | 0.81 [0.05] | 0.27 [0.18] |


Among the 100 model calibration runs, the most frequent predictions (yellow tones), approximate the 1:1 line between the
predicted and observed values. The sites with VIC lower than 50 % appear to be overestimated, and some ice-rich sites are
predicted to have lower VIC than observed (Fig. 5). Reflecting the lower prediction errors recorded with GBM, associated
predictions show a better match with the observed values (Fig. 5b). The used error distribution allows for negative (and
above 100 %VIC) values to be predicted where conditions are especially unsuitable (or suitable) for ground ice occurrence
or fall outside the range of calibration data. Thus, all negative predictions were forced to 0 %VIC and those above 100 to 100
%VIC. This had negligible effects on the evaluation statistics given that less than 0.8 % of all initial predictions from GAM
were negative and less than 0.1 % were above 100 %VIC.





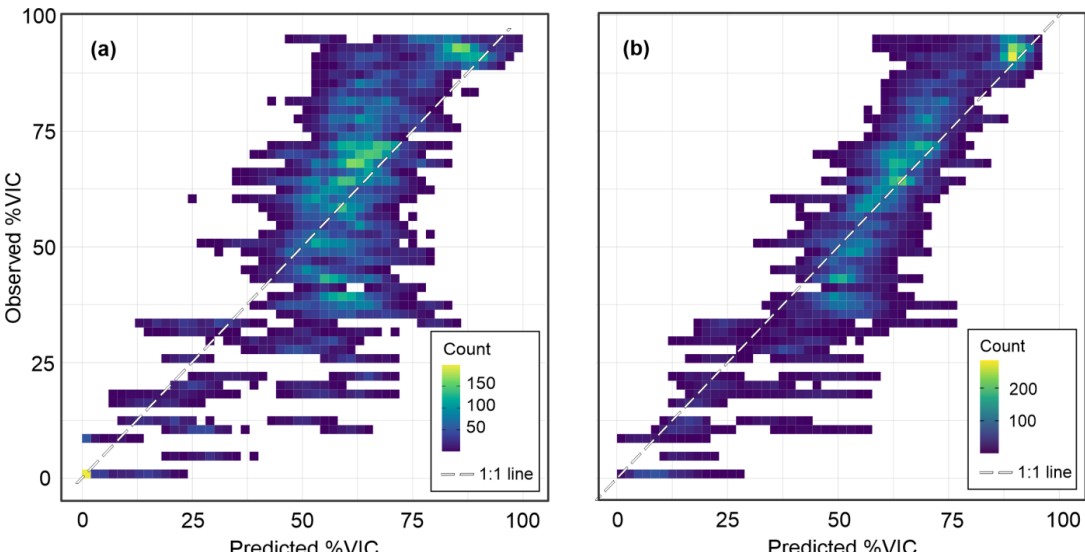


**Figure 5: Predicted volumetric ice content (VIC) plotted against observed values with generalized additive modelling (GAM) (a) and generalized boosting method (GBM) (b). Color scale represents the frequency of predicted–observed match among 100 distance-blocked cross-validation runs with calibration datasets (on average, n=251); yellow shades imply high frequency while rare occurrences are dark blue. Note different color shading for prediction frequencies between the panels.**


### 3.3 Comparisons with previous maps

The VIC predictions show a general match with the prominent high- and low-ice content areas in the reference maps but with a greater spatial variability (Fig. 6). More incongruent with the previous maps, Yedoma regions in Siberia and Alaska are predicted to have lower than expected ice content values, whereas some areas with low fine-grained sediment contents

(e.g., central Ungava Peninsula) receive higher than expected values. The reference maps each represent different measures of ground ice content: Jorgenson et al.'s (2008) map (Fig. 6b) shows total excess ice volume in the upper five meters of permafrost, while that of O'Neill et al. (2019, 2020; Wolfe et al., 2021) corresponds to the modelled segregated ice abundance in the upper five meters of permafrost (Fig. 6e), and Brown et al. (2002) estimate observation-based visible ice content in the upper 20 m of permafrost (Fig. 6c, f).


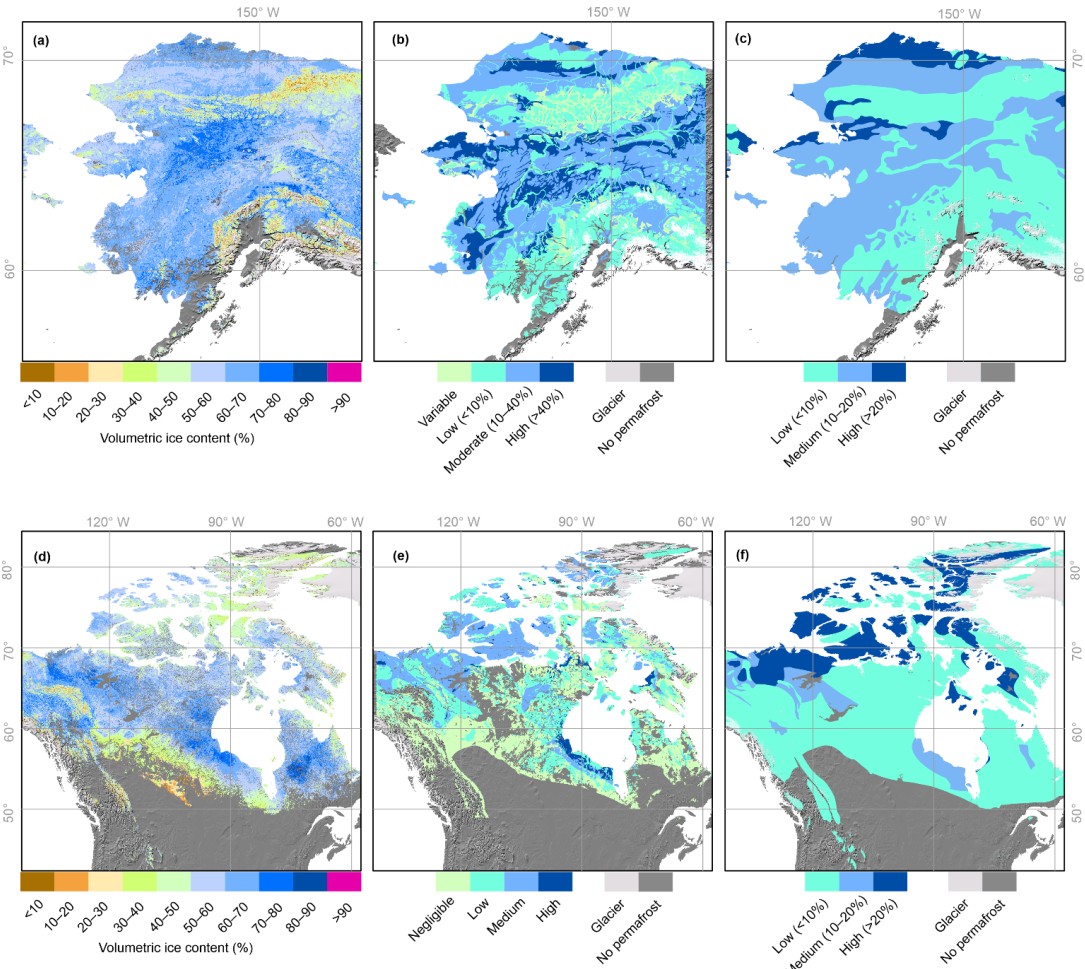

**Figure 6: Spatial comparisons of predicted ground ice content in Alaska (a, b, c) and Canada (d, e, f). The close-up maps show volumetric ice content predictions from generalized additive modelling (GAM) (a, d), excess ice volume in the upper five meters of permafrost by Jorgenson et al. (2008) (b), modelled segregated ice abundance in the upper five meters of permafrost from O'Neill et al. (2019) (e), and observation-based visible ice content in the upper 20 m of permafrost from Brown et al. (2002) (c, f). Permafrost extents are based on Obu et al. (2019) (a, d), Jorgenson et al., (2008) (b), Heginbottom et al., (1995) (e), and Brown et al. (2002) (c, f).**


## 4 Discussion

### 4.1 Spatial variability of ground ice content

Predictions of ground ice content over the Northern Hemisphere show a strong spatial variability in the topmost five meters of permafrost. Apart from distinctly different ground ice contents between lowlands and mountains, notable regional variability occurs due to soil properties and climate. Many clusters of ice-rich permafrost are associated with peatlands, but also major river basins where soil properties and topography based on the modelled responses between VIC and predictors

(Fig. C4) reflect the low-lying thick sediments with suitable conditions for ground ice aggradation. Moreover, the highest



VIC occurs in the continuous and discontinuous permafrost zones while sporadic and isolated zones often have less pronounced abundancies. We can at least partially attribute this to the climatic conditions; TDD values above ca. 1,300–1,500 °C-days are associated with sharply diminishing VIC (Fig. C4).

Using Rhorizon and fine-grained sediments as predictors was also useful in differentiating the generally thinner and coarser soils of formerly glaciated regions from thicker sediments of unglaciated regions (but also marine and fluvial sediments), and as such indirectly incorporating the influences of glacial and sedimentation history. The lower ice contents across the Canadian Shield, for example, can be largely attributed to the area's thin, glacially scoured soils (Dredge et al., 1999). On the other hand, the ice content of unglaciated regions is a result of long geomorphic and climatic history, including the complex processes behind permafrost development and global sea level changes, for example, the spatial and temporal aspects of

which cannot be captured by using data on present day conditions. Harmonized, high resolution data on past environmental conditions could improve the predictive performance of the models but applicable data were not available for this study.

**4.2 Ground ice content variability and methodological uncertainties**

Ground ice content can display high spatial variability over small horizontal distances (Wang et al., 2019; Siewert et al., 2021), but also undergo strong vertical variation within a core (Paul et al., 2020; Lacelle et al., 2022). Owing to these

characteristics, ground ice content observations have been challenging to upscale, and existing ground ice predictions remain generalized at regional to circumarctic scales. Here we computed average ground ice contents for individual grid cells often based on multiple observations (see Supplement Data 1), which is assumed to have reduced the local-scale variation between grid cells. The grid cells inheriting ground ice values from a single core are more prone to possible sampling biases, and thus potentially less representative of the ground ice conditions across the grid-cell area. Such bias could occur especially if such

a core was obtained in the area with the ice-rich permafrost. For this reason, we did not consider ice contents measured directly from ice wedges or any other massive-ice bodies (e.g., pingos). Thus, we excluded individual massive ice samples from core average computations. However, we did include ca. ten grid cells (computed based on 1–6 cores per grid cell) from palsas or lithalsas, which were considered less likely to include massive ice. By averaging the finest variation in the observation data, but also in the environmental factors (the used predictors represent average values computed for the 30 arc-

second grid cells often based on neighborhoods of varying size), statistical modelling is suited to delineate locally averaged distribution patterns in ground ice content. Although the finest variability is thus masked from the grid cell-scale predictions, the high resolution allows for discriminating, e.g., larger palsa mires from surrounding environments with lower ice content.

We aimed to maximize the comparability between observation sites by summarizing measurements in the upper five meters of permafrost, but unequally distributed sampling along the core profile could have affected site averages. That is, some

profiles only had samples from 1–2 meters below the permafrost table, which is usually the richest zone in ice content (Pollard and French, 1980; French, 2018); in many cases, occurrence of this zone is associated with the ice-rich intermediate layer of the upper permafrost (Shur, 1988; Shur et al., 2011). The highest VIC values in the modelling data from the Western Siberian lowlands (>90 %VIC, Smith et al., 2012) are mainly from the extremely ice-rich organic soils in the upper 2–3 meters of the permafrost. Predicted VIC over the region is accurately in line with these observations and has especially low

uncertainty (Fig. 4b) but the sampling depth bias could have resulted in overestimated ice contents here and in similar environments. The recorded statistically significant correlations between the depth of sampling and ground ice contents indeed imply that deeper profiles had on average lower ice contents. Moreover, the exact proportions of mineral soil and peat in the profile could not always be determined as the classification had to be made based on a varying amount of information on soil materials in the used data sources. However, if in an unlikely case any organic site was misclassified as mineral, or

vice versa, the conversion error would be around 5–15 %VIC depending on the value of GIC (Shur et al., 2021, Supporting



Information). Potential rare errors of this magnitude are argued to have negligible effect on a permafrost region-scale analysis.

Additional uncertainty for harmonizing permafrost region-wide ground ice data arises from potentially inconsistent sampling practices and documentation of ground ice content records. It is visible from the compiled data (average VIC of 61 %, Fig.

3b) that ground ice studies have been biased towards ice-rich locations (see Gilbert et al., 2016). This is because of higher research interest in ice-rich environments but also because cores from ice-poor, coarse-grained sediments or bedrock are harder to retrieve. However, some ice-rich areas were clearly less represented in the modelling datasets (Fig. A2) due to the distance-based sampling procedure (some of the clustered observations often acquired from ice-rich areas were omitted) and the number of low ice content observations in turn was relatively higher. This slightly alleviated the possible effect of biased

sampling to the modelling results.

The compiled observations represented ground ice content at one point of time, and we did not consider the potential changes in ice content over time. Whatsoever, considering that accumulation of ground ice at the targeted depth range has typically occurred during several millennia (e.g., Saito et al., 2021) we argue that any aggradation of ground ice over annual to decadal periods would have had a small effect on the average ice content across the entire depth range. For example,

Kokelj and Burn (2003) found that a decadal-scale ice aggradation of segregated ice beneath the permafrost table was limited, and that the aggraded permafrost had similar ice content to that in underlying permafrost. Permafrost degradation has resulted within many areas due to the global permafrost warming trends and widespread increase in the active-layer depths (e.g., Smith et al., 2022). However, this process is relatively slow, unless it affects massive-ice bodies (which are excluded from our estimations) and cannot significantly change average ice-content numbers calculated for a 5-m-thick

layer. Despite the discussed uncertainties, we argue that attributed to the detailed observational data and careful harmonization procedures with uncertainty estimations, the produced datasets provide a reasonably accurate account of ground ice content for modelling and validation purposes at regional and circumarctic scales.

Notwithstanding the relatively small number of ground ice observations to calibrate and evaluate the models, they performed robustly in the distance-based cross validation (i.e., evaluation statistics displayed small to moderate differences between

calibration and evaluation settings) and had moderate standard deviation in error statistics. The higher prediction uncertainty and likely overestimation of lower-end ice contents (Fig. 5) are suggested to have been due to the scarcity of observations from ice-poor landscapes (e.g., Arctic mountains ranges) and the subsequent limited extrapolation potential of the models to such areas. According to Obu et al. (2019) the extent of isolated permafrost in their zonation may be overestimated in eastern Russia and central Canada, especially. Such areas may not be underlain by permafrost and therefore associated ground ice

predictions are not applicable, but also not in the focus of the produced data. Moreover, some areas, such as parts of the central Ungava Peninsula in Canada (Fig. 4a), based on the used data have very low fine-grained sediment (sum of Clay and Silt fractions, Table 2) contents (<300 g kg$^{-1}$) that fall outside the range covered by the modelling data. As a result, associated predictions have relatively high uncertainty. These notions underline the critical need for more inclusive ground ice sampling which would benefit the extrapolation capabilities of statistical modelling outside the ice-rich environments.

**4.3 Ground ice predictions in the context of prior knowledge**

Our ground ice maps capture the general spatial patterns in ground ice distribution obtainable from previous broad-scale maps. The closest match is found outside region with Yedoma and buried glacier ice, where the pore and segregated ice account for most of the total ground ice content as demonstrated by Couture and Pollard (2017, Yukon Coastal Plain, Canada), Pollard and French (1980, Mackenzie delta, Canada), French et al. (1986, Melville Island, Canada), and Kanevskiy

et al. (2013, Beaufort Sea coast, Alaska). The most notable incongruencies are found in regions with Yedoma (Jorgenson et

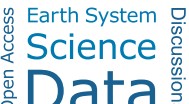

al., 2008, high ice content class in Fig. 6b; Strauss et al., 2021). Therein, soils are often very ice rich due to the presence of large syngenetic ice wedges attributing 20 to 80 % of the total volume of Yedoma, while the ice content of soils between ice wedges is highly variable, commonly ranging from 50 to 80 %VIC (Kanevskiy et al., 2011, 2012; Schirrmeister et al., 2013). Importantly, all these comparisons are markedly affected by the field observation method (e.g., visual vs. measured ground

ice estimates) and the depth for which the values are provided. Therefore, a significant level of discrepancy can be expected. Particularly we should mention the IPA map, where visible ice content estimates for the upper 20 m are inherently lower than those predicted in this study because the IPA map considered the deeper profile, but also because visible ice content does not account for pore ice that is invisible to the naked eye.

The predicted VIC values have a moderate to good match with areal VIC estimates from previous regional-scale studies. For

example, Pollard and French (1980) estimate Richard Island in Mackenzie delta to have 52 %VIC in the top 4.5 m of permafrost. In this study, the predicted mean for the same area (manually delineated in ArcGIS Pro) is 60 %VIC. Yukon Coastal Plain has been estimated to have a total ice content that ranges between 45.5 and 47.1% (Couture and Pollard, 2017) whereas here presented predictions suggest predominantly 55–65 %VIC for the region. Moreover, Couture and Pollard (2017) estimated VIC to range between 0 and 74% between the terrain units while our predictions are all >50 %VIC.

Similarly, Kanevskiy et al. (2013) concluded that the pore and segregated ice content for the upper 2–3 m of permafrost on the Beaufort Sea Coast in Alaska vary from 39 to 81 %VIC in different terrain units while here we predict VIC between 50 and 75 %. The smaller ranges can be expected owing to our approach's tendency to average ice contents to ~1 km$^2$ resolution. More spatially resolved climate and soil predictors could offer a straightforward but not easily attainable means to improve prediction accuracy of the models. Outside the ice-rich regions of the Arctic area, Wang et al. (2018) estimated an

average VIC of 31 % (±11 %) for soils 3–10 m below ground surface in the northeastern Tibetan Plateau. This matches well with our predicted VIC average of 30.9 % for the same area.

Finally, we argue that focusing on pore and segregated ice is necessary for the comparability of predicted ice content values across the permafrost domain. Spatial modelling of predominantly massive ice accumulations, such as ice wedges and buried ice, can be more readily performed with remote-sensing based object recognition techniques (Skurikhin et al., 2013; Ulrich

et al., 2014; Chen et al., 2017; Zhang et al., 2018; Witharana et al., 2020, 2021) or presence-absence modelling of landforms indicative of abundant ground ice (Karjalainen et al., 2020). We encourage different types of ground ice to be studied separately to produce tools for assessing the different thaw-related hazards associated with variable ground ice content and distribution in various permafrost environments.

**5 Data availability**

The VIC predictions and associated uncertainty layers are available at Zenodo https://doi.org/10.5281/zenodo.7009875 (Karjalainen et al., 2022). The data layers cover the Northern Hemisphere permafrost region north of the 30[th] latitude. Some areas in the northernmost Ellesmere Island and Greenland (approx. north of the 82[nd] parallel north) are clipped out due to missing NDVI data. The compiled observational ground ice data are available in the supplementary material (Supplementary

Data 1).

**6 Conclusions**

We present compilation datasets of ground ice content observations and numeric predictions of pore and segregated ground ice at 1-km resolution across the Northern Hemisphere permafrost region. The prediction maps demonstrate the substantial

spatial variation of pore and segregated ice contents in the near-surface permafrost across the region with a relatively low



prediction errors (<14 %VIC). Certain ice-rich deposits like Yedoma with abundant wedge ice, or buried ice occurrences were not explicitly included in the analysis. More accurate local variability of ground ice content could be captured by 1) incorporating additional ground ice observations, especially from areas with ice-poor permafrost and extensive regions that remain unsampled, and 2) accounting for different ground ice types. The presented observational ground ice data and

prediction maps are useful for various modelling and validation purposes at regional to circumarctic scales. The high spatial resolution also offers improved applicability for assessments of the vulnerability of natural and built environments to permafrost degradation and thaw-related hazards, hydrological processes, and biogeochemical processes involving the emissions of greenhouse gases ($CO_2$, $CH_4$, $N_2O$) from frozen soils.

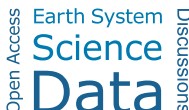

**Appendix A: Model calibration and modelling data characteristics**

Generalized additive model (GAM) was optimized by first inserting all potentially relevant environmental predictors to an initial model. Next, backward stepwise elimination was used to remove predictors from the model one by one as long as the removal decreased the model's Akaike Information Criteria (AIC) value. The model with the smallest AIC was used for

prediction. However, based on preliminary models it was observed that the predictor Rainfall caused notably weaker predictive performance and was omitted from the final modelling. Coarse sediments were excluded from the models owing to a strong bivariate correlation (>0.7 Spearman's rhoo) with Rhorizon. No exceedingly strong correlations were recorded among the predictors used in the models (each <|0.7| Spearman's rhoo, Fig. A1).

All predictors were included in the GAM model without interaction terms and assuming Gaussian error distribution with

identity link function, with maximum smoothing function set to five. The same predictors that were selected for the final GAM were also used as input for the generalized boosting method (GBM) with the following parameters: bag fraction=0.8, maximum number of trees=8,000, learning rate=0.001 and tree complexity=5. Rather large bag fraction (80 % of observations used for random samples for each individual tree) was selected due to the relatively small calibration data. Maximum number of trees, learning rate and tree complexity were calibrated as such that the models reach minimum

predictive error using at least 1,000 trees (Elith et al., 2008). Notwithstanding, a slow learning rate was selected to diminish the effect of potentially spurious predictions from individual trees (owing to the relatively small size of the calibration datasets). As a result, from 3,500 to 8,000 trees were fitted depending on the cross-validation run to optimize the model by *gbm.step* function in package dismo (Hijmans et al., 2017).

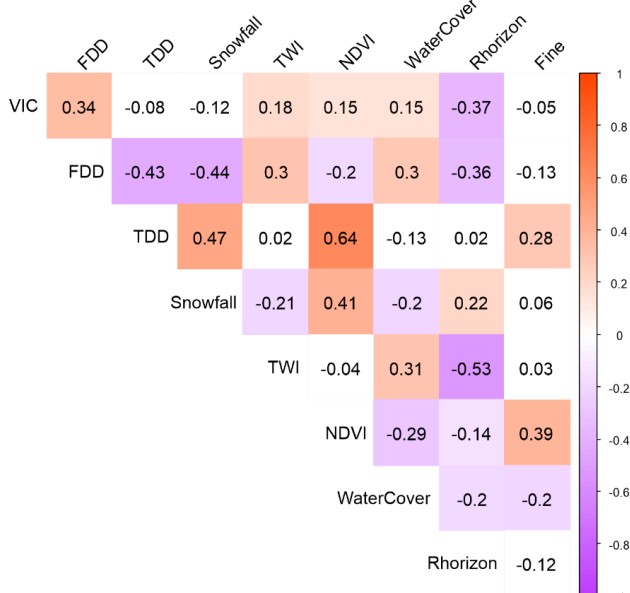


**Figure A1: Bivariate Spearman's rank order correlations between the environmental predictors that were included in the final models. Positive (orange) and negative (purple) background stand for correlations according to their strength in volumetric ice content (VIC) modelling. Non-significant correlations at a 0.01 confidence level are presented with white background. Abbreviated predictors are freezing- and thawing degree-days (FDD and TDD), topographic wetness index (TWI), Normalized Difference**

**Vegetation Index (NDVI), coverage of open water bodies (WaterCover), occurrence probability of R horizon within 200 cm from the surface (Rhorizon), and fine sediment proportions (Fine).**





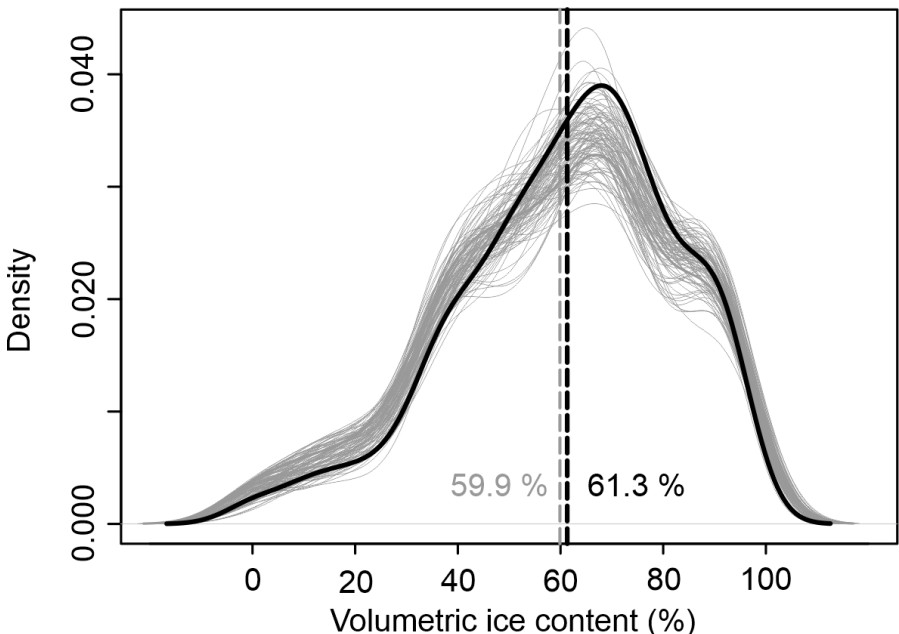

**Figure A2: Density distributions for the compiled ground ice content datasets and the model calibration samples. In total, 437 volumetric ice content observations denoted with black density distribution curve are used in the modelling. The grey curves show the density distributions for the 100 random calibration samples that were made spatially independent by applying geostatistically estimated threshold for spatial autocorrelation. Vertical dashed lines show mean values for the compiled ground ice observations (black) and mean among 100 calibration datasets (grey).**




**Appendix B: Analysis of spatial autocorrelation**

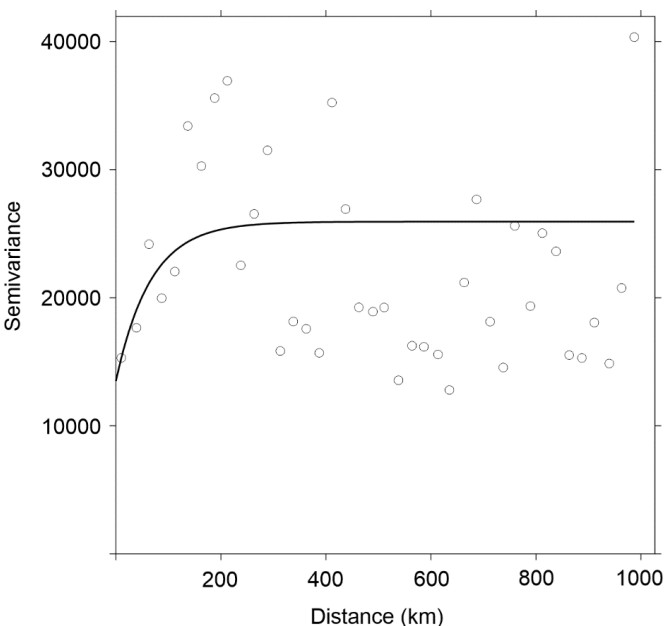


**Figure B3: Exponential variogram model that was used to assess the effect of spatial autocorrelation. The model was fitted for the residuals from generalized additive models for volumetric ice content observations using a bin width of 25 km and a cutoff value of 1,000 km.**



**Appendix C: Modelled responses between volumetric ice content and environmental predictors**

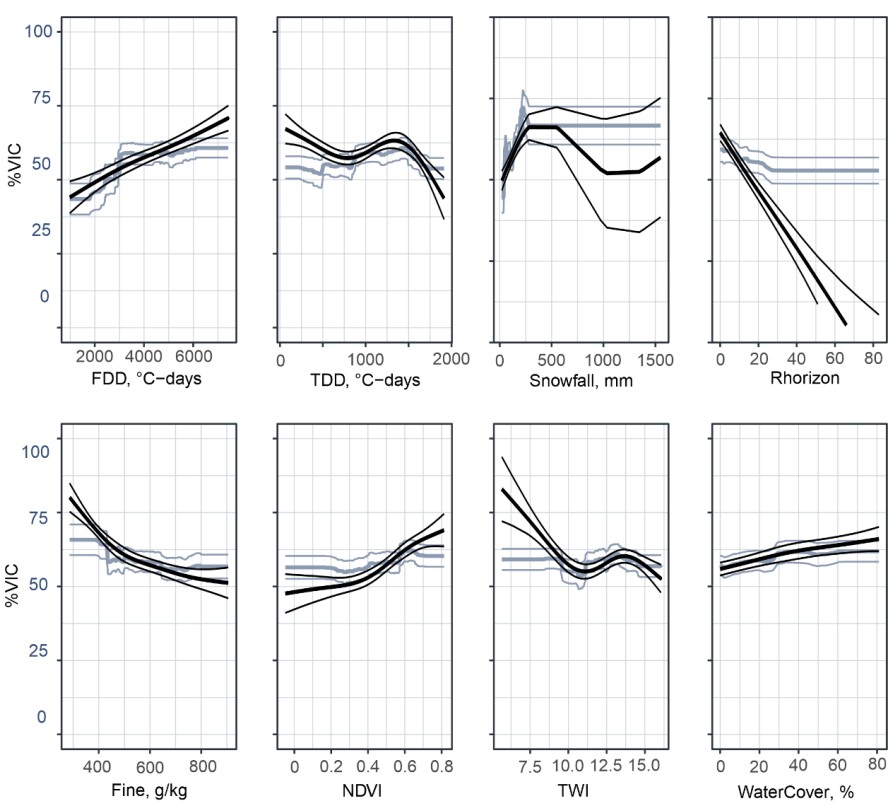

**Figure C4: Averaged response curves from 100 model realizations. Black lines correspond to the results from generalized additive**

**modelling ja and grey to generalized boosting method. The curves show the shape of the effects of predictor variables to**

**volumetric ice content (%VIC) over the range of their values. Thick line is the mean from 100 distance-based cross-validations and**

**thin lines show 1 standard deviation.**





**Appendix D: Spatial predictions**


**Figure D5: Predicted volumetric ice content (VIC, %) based on the generalized additive model for North America (a) and Eurasia (b). Locations mentioned in text are numbered: 1) Lena River, 2) Kolyma River, 3) Yenisei River, 4) Yukon River, 5) Mackenzie River, 6) Western Siberia peatlands, 7) Hudson Bay peatlands, and 8) the Tibetan Plateau, and 9) the central Ungava Peninsula. Permafrost extent is based on Obu et al., (2019).**






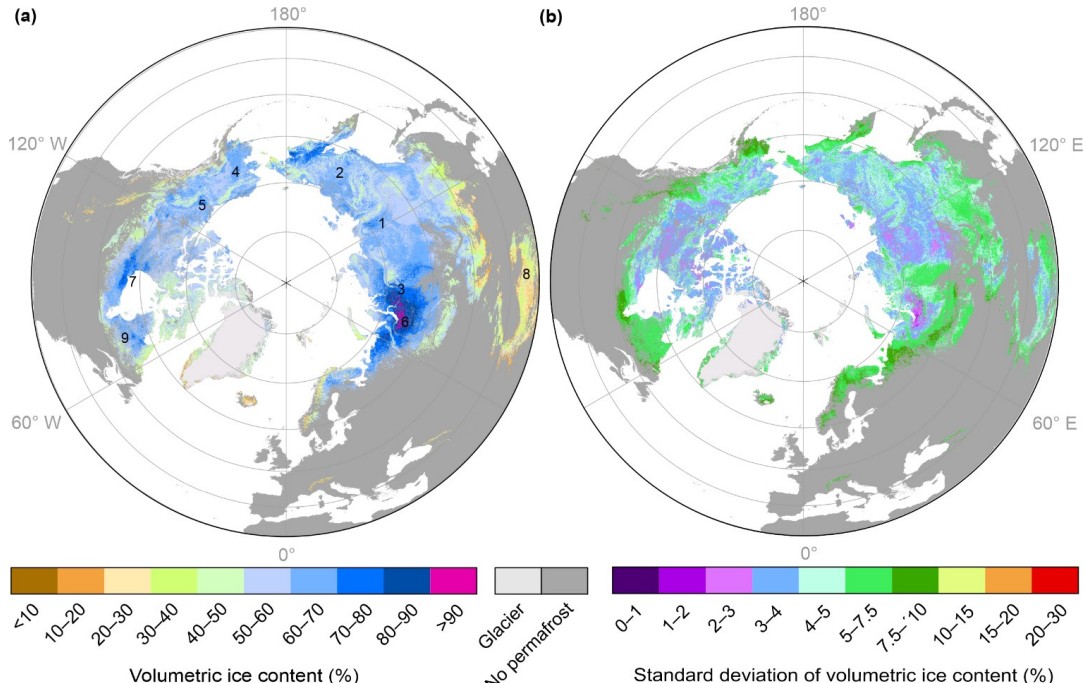

**Figure D6: Predicted volumetric ice content (a, %VIC) and their standard deviation (b) for the Northern Hemisphere permafrost region based on 100 generalized boosting method rounds. Locations mentioned in text are numbered: 1) Lena River, 2) Kolyma River, 3) Yenisei River, 4) Yukon River, 5) Mackenzie River, 6) West Siberian Lowland, 7) Hudson Bay Lowland, 8) the Tibetan Plateau, and 9) the central Ungava Peninsula. Permafrost extent is based on Obu et al. (2019).**




**Figure D7: Predicted volumetric ice content (VIC, %) based on the generalized boosting method for North America (a) and Eurasia (b). Locations mentioned in text are numbered: 1) Lena River, 2) Kolyma River, 3) Yenisei River, 4) Yukon River, 5) Mackenzie River, 6) Western Siberia peatlands, 7) Hudson Bay peatlands, 8) the Tibetan Plateau, and 9) the central Ungava Peninsula. Permafrost extent is based on Obu et al., (2019).**





**Appendix E: Ground ice data sources**


Abramov, A., Gruber, S. and Gilichinsky, D.: Mountain permafrost on active volcanoes: field data and statistical mapping, Klyuchevskaya volcano group, Kamchatka, Russia, Permafrost Periglac. Process., 19, 261–277, https://doi.org/10.1002/ppp.622, 2008.

ADAPT: Cryostratigraphy, carbon and nitrogen content and 14C dating of permafrost cores from sites across the Canadian Arctic, v. 1.0 (2013-2014), Nordicana D25, https://doi.org/10.5885/45427AD-06F05740704B4CA3, 2016.

Alexeev, S., Arzhannikov, S. and Alexeeva, L.: The evolution of permafrost in the western part of the Todzha depression, Russia, Geomorphology, 91, 124–131, https://doi.org/10.1016/j.geomorph.2007.02.001, 2007.

Allard, M., Caron, S. and Begin, Y.:Climatic and ecological controls on ice segregation and thermokarst: the case history of a permafrost plateau in Northern Quebec, Permafrost Periglac. Process., 7, 207–227, 1996.

Benkert, B., Fortier, D., Kennedy, K. and Lewkowicz, A. G.: Northern Climate ExChange. Burwash Landing and Destruction Bay Landscape Hazards:
Geological Mapping for Climate Change Adaptation Planning, Yukon Research Centre, Yukon College, 111 pp. and 2 maps, 2013.

Benkert, B. E., Fortier, D., Lipovsky, P., Lewkowicz, A., Roy, L.-P., de Grandpré, I., Grandmont, K., Turner, D., Laxton, S. and Moote, K.: Faro Landscape Hazards: Geoscience Mapping for Climate Change Adaptation Planning, Northern Climate ExChange, Yukon Research Centre, Yukon College, 130 pp. and 2 maps, 2015a.

Benkert, B. E., Fortier, D., Lipovsky, P., Lewkowicz, A., de Grandpré, I., Grandmont, K., Turner, D., Laxton, S., Moote, K. and Roy, L.-P.: Ross River
Landscape Hazards: Geoscience Mapping for Climate Change Adaptation Planning, Northern Climate ExChange, Yukon Research Centre, Yukon College. 116 pp. and 2 maps, 2015b.

Benkert, B. E., Kennedy, K., Fortier, D., Lewkowicz, A., Roy, L.-P., Grandmont, K., de Grandpré, I., Laxton, S., McKenna, K. and Moote, K.: Dawson City Landscape Hazards: Geoscience Mapping for Climate Change Adaptation Planning. Northern Climate ExChange, Yukon Research Centre, Yukon College. 166 pp. and 2 maps, 2015c.

Benkert, B. E., Kennedy, K., Fortier, D., Lewkowicz, A., Roy, L.-P., de Grandpré, I., Grandmont, K., Drukis, S., Colpron, M., Light, E. and Williams, T.: Old Crow landscape hazards: Geoscience mapping for climate change adaptation planning. Northern Climate ExChange, Yukon Research Centre, Yukon College. 136 pp. and 2 maps, 2016.

Bjella, K. L., Shur, Y., Kanevskiy, M., Duvoy, P., Grunau, B., Best, J., Bourne, S. and Affleck, R. T.: Improving design methodologies and assessment tools for building on permafrost in a warming climate, Technical Report (Engineer Research and Development Center (U.S.)), ERDC/CRREL TR-20-13,
220 pp., http://dx.doi.org/10.21079/11681/38879, 2020.

Blanchet, J. and Davison, A. C.: Statistical modelling of ground temperature in mountain permafrost, P. Roy. Soc. A-Math. Phy., 468, 1472–1495, https://doi.org/10.1098/rspa.2011.0615, 2012.

Budantseva, N. A., Gorshkov, E. I., Isaev, V. S., Semenkov, I. N., Usov, A. N., Chizhova, Ju. N. and Vasil'chuk, Yu. K.: Engineering-geological and geochemical features of palsa and lithalsa landscapes in the area of the Khanovey Science Education Station, Inzhenernaya Geologiya, 3/2015, 34–82, 2015.

Calmels, F. and Allard, M.: Segregated ice structures in various heaved permafrost landforms through CT scan, Earth Surf. Proc. Land., 33, 209–225, https://doi.org/10.1002/esp.1538, 2008.

Chen, J., Zhao, L., Sheng, Y., Li, J., Wu, X., Du, E., Liu, G. and Pang, Q.: Some characteristics of permafrost and its distribution in the Gaize area on the Qinghai—Tibet Plateau, China, Arct. Antarct. Alp. Res., 48, 395–409, http://dx.doi.org/10.1657/AAAR0014-023, 2016.

Coultish, T. L. and Lewkowicz, A. G.: Palsa dynamics in a subarctic mountainous environment, Wolf Creek, Yukon Territory, Canada. Proceedings of the
Eighth International Conference on Permafrost, International Conference on Permafrost, Proceedings, 8, 163–168, 2003.

Ensom, T., Morse, P. D., Kokelj, S. V., MacDonald, E., Young, J., Tank, S., Subedi, R., Grozic, E. and Gastagner, A.: Permafrost geotechnical borehole data synthesis: 2013-2017 Inuvik-Tuktoyaktuk region, Northwest Territories, Geological Survey of Canada, Open File 8652, 78 pp., https://doi.org/10.4095/321869, 2020.

Fontaine, M.: Ground Ice Content and Geochemistry of Active Layer and Permafrost in Northwestern Arctic Canada, M.S. thesis, University of Ottawa,
Ottawa, Canada, 87 pp., http://dx.doi.org/10.20381/ruor-4005, 2016.



Fortier, D., Allard, M. and Shur, Y.: Observation of Rapid Drainage System Development by Thermal Erosion of Ice Wedges on Bylot Island, Canadian Arctic Archipelago, Permafrost Periglac. Process., 18, 229–243, https://doi.org/10.1002/ppp.595, 2007.

Frappier, R. and Lacelle, D.: Distribution, morphometry, and ice content of ice-wedge polygons in Tombstone Territorial Park, central Yukon, Canada, Permafrost Periglac. Process., 32, 587–600, https://doi.org/10.1002/ppp.2123, 2021.

French, H. M., Bennett, L. and Hayley, D. W.: Ground ice conditions near Rea Point and on Sabine Peninsula, eastern Melville Island, Can. J. Earth Sci., 23, 1389–1400, 1986.

Fritz, M., Wetterich, S., Schirrmeister, L., Meyer, H., Lantuit, H., Preusser, F. and Pollard, W. H.: Eastern Beringia and beyond: late Wisconsinan and Holocene landscape dynamics along the Yukon Coastal Plain, Canada, Palaeogeogr. Palaeocl, 319, 28–45, https://doi.org/10.1016/j.palaeo.2011.12.015, 2012.

Fuchs, M., Grosse, G., Jones, B. M., Strauss, J., Baughman, C. A. and Walker, D. A.: Sedimentary and geochemical characteristics of two small permafrost-dominated Arctic river deltas in northern Alaska, arktos - The Journal of Arctic Geosciences, 4, 1–18, https://doi.org/10.1007/s41063-018-0056-9, 2018.

Fuchs, M., Lenz, J., Jock, S., Nitze, I., Jones, B. M., Strauss, J., Günther, F. and Grosse, G.: Basic sediment characteristics of permafrost cores in the Teshekpuk Lake Area on the Arctic Coastal Plain, Northern Alaska, PANGAEA, https://doi.org/10.1594/PANGAEA.895163, 2018.

Gagnon, S. and Allard, M.: Ground surface temperatures, soil properties and vegetation cover in the Narsajuaq River Valley, Nunavik, Canada, v. 1.0 595 (2016-2018), Nordicana D76, https://doi.org/10.5885/45657CE-4AA8A7A640934B9E, 2020.

Gilbert, G. L., Cable, S., Thiel, C., Christiansen, H. H. and Elberling, B.: Cryostratigraphy, sedimentology, and the late Quaternary evolution of the Zackenberg River delta, northeast Greenland, Cryosphere, 11, 1265–1282, https://doi.org/10.5194/tc-11-1265-2017, 2017.

Heslop, J. K., Chandra, S., Sobzcak, W. V., Davydov, S. P., Davydova, A. I., Spektor, V. V. and Walter Anthony, K. M.: Variable respiration rates of incubated permafrost soil extracts from the Kolyma River lowlands, north-east Siberia, Polar Res., 36, 1305157, 600 https://doi.org/10.1080/17518369.2017.1305157, 2017.

Holloway, J.: Impacts of Forest Fire on Permafrost in the Discontinuous Zones of Northwestern Canada, Ph.D. thesis, University of Ottawa, Ottawa, Canada, 212 pp., http://dx.doi.org/10.20381/ruor-25410, 2020.

Ingeman-Nielsen, T.: Oedometer tests on frozen cores from Qaanaaq, Greenland, Technical University of Denmark, Department of Civil Engineering, BYG R-447, 41 pp., https://orbit.dtu.dk/en/publications/72cb7311-babe-4ac3-92d2-03847e2f0579, 2020.

Iwahana, G., Machimura, T., Kobayashi, Y., Fedorov, A. N., Konstantinov, P. Y. and Fukuda, M.: Influence of forest clear-cutting on the thermal and hydrological regime of the active layer near Yakutsk, eastern Siberia, J. Geophys. Res.-Biogeo., 110, G02004, https://doi.org/10.1029/2005JG000039, 2005.

Iwahana, G., Fukui, K., Mikhailov, N., Ostanin, O. and Fujii, Y.: Internal structure of a lithalsa in the Akkol Valley, Russian Altai Mountains. Permafrost Periglac. Process., 23, 107–118, https://doi.org/10.1002/ppp.1734, 2012.

Iwahana, G., Takano, S., Petrov, R. E., Tei, S., Shingubara, R., Maximov, T. C., Fedorov, A. N., Desyatkin, A. R., Nikolaev, A. N., Desyatkin, R. V. and 610 Sugimoto, A.: Geocryological characteristics of the upper permafrost in a tundra-forest transition of the Indigirka River Valley, Russia, Polar Sci., 8, 96–113, http://dx.doi.org/10.1016/j.polar.2014.01.005, 2014.

Iwahana, G., Harada, K., Uchida, M., Tsuyuzaki, S., Saito, K., Narita, K., Kushida, K. and Hinzman, L. D.: Geomorphological and geochemistry changes in permafrost after the 2002 tundra wildfire in Kougarok, Seward Peninsula, Alaska, J. Geophys. Res.-Earth., 121, 1697–1715, https://doi.org/10.1002/2016JF003921, 2016.

Jongejans, L. L., Strauss, J., Lenz, J., Peterse, F., Mangelsdorf, K., Fuchs, M. and Grosse, G.: Organic matter characteristics in Yedoma and thermokarst deposits on Baldwin Peninsula, west Alaska, Biogeosciences, 15, 6033–6048, https://doi.org/10.5194/bg-15-6033-2018, 2018.

Jorgenson, T.: Permafrost soil database with information on site, topography, geomorphology, hydrology, soil stratigraphy, soil carbon, ground ice isotopes, and vegetation at thermokarst features near Toolik and Noatak River, 2009-2013, Arctic Data Center, https://doi.org/10.18739/A2CP5C, 2013.

Jorgenson, M. T., Harden, J., Kanevskiy, M., O'Donnell, J., Wickland, K., Ewing, S., Manies, K., Zhuang, Q., Shur, Y. and Striegl, R.: Reorganization of 620 vegetation, hydrology and soil carbon after permafrost degradation across heterogeneous boreal landscapes, Environ. Res. Lett., 8, 035017, https://doi.org/10.1088/1748-9326/8/3/035017, 2013.

Jorgenson, T.: Permafrost soil database with information on site, topography, geomorphology, hydrology, soil stratigraphy, soil carbon, ground ice isotopes, and vegetation at thermokarst features near Toolik and Noatak River, 2009-2013, Arctic Data Center, https://doi.org/10.18739/A2CP5C, 2015.



Kanevskiy, M., Shur, Y., Fortier, D., Jorgenson, M. T. and Stephani, E.: Cryostratigraphy of late Pleistocene syngenetic permafrost (Yedoma) in northern Alaska, Itkillik River exposure, Quaternary Res., 75, 584–596, https://doi.org/10.1016/j.yqres.2010.12.003, 2011.

Kanevskiy, M., Shur, Y., Connor, B., Dillon, M., Stephani, E. and O'Donnell, J.: Study of ice-rich syngenetic permafrost for road design (Interior Alaska), Proceedings Tenth International Conference on Permafrost, Vol. 1, 25–29, 2012.

Kanevskiy, M., Shur, Y., Krzewinski, T. and Dillon, M.: Structure and properties of ice-rich permafrost near Anchorage, Alaska, Cold Reg. Sci. Technol., 93, 1–11, https://doi.org/10.1016/j.coldregions.2013.05.001, 2013a.

Kanevskiy, M., Shur, Y., Jorgenson, M., Ping, C.-L., Michaelson, G., Fortier, D., Stephani, E., Dillon, M. and Tumskoy, V.: Ground ice in the upper permafrost of the Beaufort Sea coast of Alaska, Cold Reg. Sci. Technol., 85, 56–70, https://doi.org/10.1016/j.coldregions.2012.08.002, 2013b.

Kanevskiy, M., Jorgenson, T., Shur, Y., O'Donnell, J. A., Harden, J. W., Zhuang, Q. and Fortier, D.: Cryostratigraphy and permafrost evolution in the lacustrine lowlands of west-central Alaska, Permafrost Periglac. Process., 25, 14–34, https://doi.org/10.1002/ppp.1800, 2014.

Kanevskiy, M. and Jorgenson, M. T.: Ground ice data, Horseshoe Lake, Arctic Data Center, https://doi.org/10.18739/A2NW69, 2015.

Kanevskiy, M. Jorgenson, M. T. and Shur, Y.: Ground ice data, Creamer's Field, Arctic Data Center, https://doi.org/10.18739/A26Q08, 2015.

Kanevskiy, M., Shur, Y., Jorgenson, T., Brown, D. R. N., Moskalenko, N., Brown, J., Walker, D. A., Raynolds, M. K. and Buchhorn, M.: Degradation and stabilization of ice wedges: implications for assessing risk of thermokarst in northern Alaska, Geomorphology, 297, 20–42, https://doi.org/10.1016/j.geomorph.2017.09.001, 2017.

Kanevskiy, M., Jones, B. and Arp, C.: Cryostratigraphy and ground-ice content of the upper permafrost in drained-lake basins, northern Alaska, April-May
2019, Arctic Data Center, https://doi.org/10.18739/A2KP7TS1Q, 2020a.

Kanevskiy, M., Jorgenson, T., Liljedahl, A. and Daanen, R. P.: Cryostratigraphy and ground-ice content of the upper permafrost at the Jago River study site, Northern Alaska, July-August 2018, Arctic Data Center, https://doi.org/10.18739/A22J6853K, 2020b.

Kenner, R., Noetzli, J., Hoelzle, M., Raetzo, H. and Phillips, M.: Distinguishing ice-rich and ice-poor permafrost to map ground temperatures and ground ice occurrence in the Swiss Alps, Cryosphere, 13, 1925–1941, https://doi.org/10.5194/tc-13-1925-2019, 2019.

Kokelj, S. V., Smith, C. A. S. and Burn, C. R.: Physical and chemical characteristics of the active layer and permafrost, Herschel Island, western Arctic Coast, Canada, Permafrost Periglac. Process., 13, 171–185, https://doi.org/10.1002/ppp.417, 2002.

Kukkonen, I. T., Suhonen, E., Ezhova, E., Lappalainen, H., Gennadinik, V., Ponomareva, O., Gravis, A., Miles, V., Kulmala, M., Melnikov, V. and Drozdov, D.: Observations and modelling of ground temperature evolution in the discontinuous permafrost zone in Nadym, north-west Siberia, Permafrost Periglac. Process., 31, 264–280, https://doi.org/10.1002/ppp.2040, 2020.

Le, T. M. H., Depina, I., Guegan, E. and Sinitsyn, A.: Thermal regime of permafrost at Varandey Settlement along the Barents Sea Coast, North West Arctic Russia, Eng. Geol., 246, 69–81, https://doi.org/10.1016/j.enggeo.2018.09.026, 2018.

Li, D., Chen, J., Meng, Q., Liu, D., Fang, J. and Liu, J.: Numeric simulation of permafrost degradation in the Eastern Tibetan Plateau, Permafrost Periglac. Process., 19, 93–99, https://doi.org/10.1002/ppp.611, 2008.

Lin, Z., Gao, Z., Fan, X., Niu, F., Luo, J., Yin, G. and Liu, M.: Factors controlling near surface ground-ice characteristics in a region of warm permafrost,
Beiluhe Basin, Qinghai-Tibet Plateau, Geoderma, 376, 114540, https://doi.org/10.1016/j.geoderma.2020.114540, 2020.

L'Hérault, E.: Contexte climatique critique favorable au déclenchement de ruptures de mollisol dans la vallée de Salluit, Nunavik, M.S. thesis, 138 pp., http://hdl.handle.net/20.500.11794/21909, 2009 (In French).

L'Hérault E., Allard, M., Fortier, D., Carbonneau, A.-S., Doyon-Robitaille, J., Lachance, M.-P., Ducharme, M-A., Larrivée, K., Grandmont, K. and Lemieux, C.: Production de cartes prédictives des caractéristiques du pergélisol afin de guider le développement de l'environnement bâti pour quatre
communautés du Nunavik, Rapport final, Québec, Centre d'études nordiques, Université Laval, 90 pp., 2013 (In French).

Ludwig, S., Holmes, R., Natali, S., Mann, P., Schade, J. and Jardine, L.: Polaris Project 2017: Permafrost carbon and nitrogen, Yukon-Kuskokwim Delta, Alaska, Arctic Data Center, https://doi.org/10.18739/A2ZG6G72V, 2018.

Maslakov, A. A., Egorov, E. G. and Zelensky, G. M.: Permafrost transient layer of Eastern Chukotka Coastal Plains, Рельеф и четвертичные образования Арктики, Субарктики и Северо-Запада России, 7, 118–123, https://doi.org/10.24411/2687-1092-2020-10717, 2020 (In Russian).





Meyer, H., Schirrmeister, L., Andreev, A., Wagner, D., Hubberten, H.-W., Yoshikawa, K., Bobrov, A., Wetterich, S. Opel, T., Kandiano, E. and Brown, J.: Lateglacial and Holocene isotopic and environmental history of northern coastal Alaska–Results from a buried ice-wedge system at Barrow, Quaternary Sci. Rev., 29, 3720–3735, https://doi.org/10.1016/j.quascirev.2010.08.005, 2010.

Michaelson, G. J., Ping, C. L., Epstein, H., Kimble, J. M. and Walker, D. A.: Soils and frost boil ecosystems across the North American Arctic Transect, J. Geophys. Res., 113, G03S11, https://doi.org/10.1029/2007JG000672, 2008

Munroe, J. S., Doolittle, J. A., Kanevskiy, M. Z., Hinkel, K. M., Nelson, F. E., Jones, B. M., Shur, Y. and Kimble, J. M.: Application of ground-penetrating radar imagery for three-dimensional visualisation of near-surface structures in ice-rich permafrost, Barrow, Alaska, Permafrost Periglac. Process., 18, 309–321, https://doi.org/10.1002/ppp.594, 2007.

Oblogov, G. E., Vasiliev, A. A., Streletskaya, I. D., Zadorozhnaya, N. A., Kuznetsova, A. O., Kanevskiy, M. Z. and Semenov, P. B.: Methane content and emission in the permafrost landscapes of Western Yamal, Russian Arctic, Geosciences, 10, 412, https://doi.org/10.3390/geosciences10100412, 2020.

Oliver, L. K.: Characterization of permafrost development by isotopic and chemical analysis of soil cores taken from the Copper River Basin and an upland loess deposit in interior Alaska, Ph.D. thesis, University of Alaska Fairbanks, 195 pp., http://hdl.handle.net/11122/9134, 2012.

Osterkamp, T. E., Jorgenson, M. T., Schuur, E. A. G., Shur, Y. L., Kanevskiy, M. Z., Vogel, J. G. and Tumskoy, V. E.: Physical and ecological changes associated with warming permafrost and thermokarst in interior Alaska, Permafrost Periglac. Process., 20, https://doi.org/10.1002/ppp.1952, 2009.

O'Neill, H. B. and Burn, C. R.: Physical and temporal factors controlling the development of near-surface ground ice at Illisarvik, western Arctic coast,
Canada, Can. J. Earth Sci., 49, 1096–1110 https://doi.org/10.1139/e2012-043, 2012.

Paquette, M., Rudy, A. C. A., Fortier, D. and Lamoureux, S. F.: Multi-scale site evaluation of a relict active layer detachment in a High Arctic landscape, Geomorphology, 359, 107159, https://doi.org/10.1016/j.geomorph.2020.107159, 2020.

Paul, J. R., Kokelj, S. V. and Baltzer, J. L.: Spatial and stratigraphic variation of near-surface ground ice in discontinuous permafrost of the taiga shield, Permafrost Periglac. Process., 32, 3–18, https;//doi.org/10.1002/ppp.2085, 2020.

Pumple, J., Froese, D. and Calmels, F.: Characterizing permafrost valley fills along the Alaska Highway, southwest Yukon, Proceedings GeoQuébec 2015 68th Canadian Geotechnical Conference and 7th Canadian Permafrost Conference, 20–23, 2015.

Pumple, J. D.: Characterizing permafrost along the Alaska Highway, Southwestern Yukon, Canada, M.S. thesis, University of Alberta, Edmonton, Canada, 212 pp., https://doi.org/10.7939/R30Z71716, 2016.

Rasmussen, L. H., Zhang, W., Hollesen, J., Cable, S., Christiansen, H. H., Jansson, P.-E. and Elberling, B.: Modelling present and future permafrost thermal
regimes in Northeast Greenland, Cold Reg. Sci. Technol., 146, 199–2013, https://doi.org/10.1016/j.coldregions.2017.10.011, 2018.

Roy, C.: The origin of massive ground ice in raised marine sediments along the Eureka Sound Lowlands, Nunavut, Canada, M.S. Thesis, McGill University, Montréal, Canada, 122 pp., 2018.

Schirrmeister, L., Grosse, G., Kunitsky, V., Meyer, H., Derivyagin, A. and Kuznetsova, T.: Permafrost, periglacial and paleo-environmental studies on New Siberian Islands, in: Russian Polar Research, 195–314, AWI, Alfred-Wegener-Institut für Polar-und Meeresforschung,
https://oceanrep.geomar.de/id/eprint/28041, 2003.

Schirrmeister, L., Wagner, D., Grigoriev, M. and Bolshiyanov, D.: The expedition LENA 2005, in: Expeditions in Siberia in 2005, edited by: Schirrmeister, L., Berichte zur Polar-und Meeresforschung (Reports on Polar and Marine Research), 550, 41–242, https://oceanrep.geomar.de/id/eprint/28054, 2007.

Schirrmeister, L., Grosse, G., Kunitsky, V., Magens, D., Meyer, H., Dereviagin, A., Kuznetsova, T., Andreev, A., Babiy, O., Kienast, F., Grigoriev, M., Overduin, P. P. and Preusser, F.: Periglacial landscape evolution and environmental changes of Arctic lowland areas for the last 60 000 years (western
Laptev Sea coast, Cape Mamontov Klyk), Polar Res., 27, 249–272, https://doi.org/10.1111/j.1751-8369.2008.00067.x, 2008.

Schirrmeister, L., Pestryakova, L., Wetterich, S. and Tumskoy, V.: Joint Russian-German polygon project: East Siberia 2011 - 2014; the expedition Kytalyk 2011, Berichte zur Polar- und Meeresforschung, (Reports on polar and marine research), Bremerhaven, Alfred Wegener Institute for Polar and Marine Research, 653, 153 pp., https://doi.org/10.2312/BzPM_0653_2012, 2012.

Schirrmeister, L., Grosse, G., Kunitsky, V. V. and Siegert, C.: Sedimentological, biogeochemical and geochronological data from permafrost deposit
Kurungnakh, PANGAEA, https://doi.pangaea.de/10.1594/PANGAEA.884069, 2017a.

Schirrmeister, L., Grosse, G., Kunitsky, V. V. and Siegert, C.: Sedimentological, biogeochemical and geochronological data from permafrost deposit Nagym, PANGAEA, https://doi.org/10.1594/PANGAEA.884063, 2017b.





Schirrmeister, L., Bobrov, A. A., Raschke, E. and Wetterich, S.: Sediment, ground ice, geochronological and paleoecological data from polygon cores in the Siberian Arctic, PANGAEA, https://doi.org/10.1594/PANGAEA.887933, 2018.

Schirrmeister, L., Dietze, E., Matthes, H., Grosse, G., Strauss, J., Laboor, S., Ulrich, M., Kienast, F. and Wetterich, S.: The genesis of Yedoma Ice Complex permafrost – grain-size endmember modeling analysis from Siberia and Alaska, E&G Quaternary Science Journal, 69, 33–53, https://doi.org/10.5194/egqsj-69-33-2020, 2020.

Schwamborn, G., Meyer, H., Fedorov, G., Schirrmeister, L. and Hubberten, H. W.: Ground ice and slope sediments archiving late Quaternary paleoenvironment and paleoclimate signals at the margins of El'gygytgyn Impact Crater, NE Siberia, Quaternary Res., 66, 259–272,
https://doi.org/10.1016/j.yqres.2006.06.007, 2006.

Schwamborn, G., Heinzel, J. and Schirrmeister, L: Internal characteristics of ice-marginal sediments deduced from georadar profiling and sediment properties (Brøgger Peninsula, Svalbard), Geomorphology, 95, 74–83, https://doi.org/10.1016/j.geomorph.2006.07.032, 2008.

Seguin, M.-K. and Frydecki, J.:Semi-quantitative geophysical investigation of permafrost in northern Québec, J. Appl. Geophys., 32, 73–84, https://doi.org/10.1016/0926-9851(94)90010-8, 1994.

Seppälä, M., Gray, J. and Ricard, J.: Development of low–centred ice–wedge polygons in the northernmost Ungava Peninsula, Québec, Canada, Boreas, 20, 259–285, https://doi.org/10.1111/j.1502-3885.1991.tb00155.x, 1991.

Sharkhuu, A., Sharkhuu, N., Etzelmüller, B, Flo Heggem, E. S., Nelson, F. E., Shiklomanov, N. I., Goulden, C. E. and Brown, J.: Permafrost monitoring in the Hovsgol mountain region, Mongolia, J. Geophys. Res., 112, F02S06, https://doi.org/10.1029/2006JF000543, 2007.

Shur, Y., Kanevskiy, M., Dillon, M., Stephani, E. and O'Donnell, J.: Geotechnical investigations for the Dalton Highway innovation project as a case study
of the ice-rich syngenetic permafrost, Alaska University Transportation Center, Alaska Department of Transportation and Public Facilities, 156 pp., 2010.

Shur, Y., Kanevskiy, M., Jorgenson, M., Dillon, M., Stephani, E., Bray, M. and Fortier, D.: Permafrost Degradation and Thaw Settlement under Lakes in Yedoma Environment, in: Proceedings of the Tenth international conference on Permafrost, 383–388, 2012.

Slagoda, E. A., Leibman, M. O., Khomutov, A. V. and Orekhov, P. T.: Cryolithologic construction of the first terrace at Bely Island, Kara Sea (Part 1), Earth's Cryosphere, 17, 11–21, 2013.

Smith, L. C., Beilman, D. W., Kremenetski, K. V., Sheng, Y., MacDonald, G. M., Lammers, R. B., Shiklomanov, A. I. and Lapshina, E. D.: Short Communication: Influence of permafrost on water storage in West Siberian peatlands revealed from a new database of soil properties, Permafrost. Periglac. Processes 23, 69–79, https://doi.org/10.1002/ppp.735, 2012.

Smith, S. L., Chartrand, T. N., Nguyen, T.-N., Riseborough, D. W., Ednie, M. and Ye, S.: Geotechnical Database and Descriptions of Permafrost Monitoring Sites Established 2006-07 in the Central and Southern Mackenzie Corridor, Geological Survey of Canada, Open File 6041, 188 pp.,
https://doi.org/10.4095/226435, 2009.

Smith, S. L., Riseborough, D. W. and Bonnaventure, P. P.: Eighteen year record of forest fire effects on ground thermal regimes and permafrost in the Central Mackenzie Valley, NWT, Canada, Permafrost Periglac. Process., 26, 289–303, https://doi.org/10.1002/ppp.1849, 2015.

Stephani, E., Fortier, D. and Shur, Y. L.: Applications of cryofacies approach to frozen ground engineering – Case study of a road test site along the Alaska Highway (Beaver Creek, Yukon, Canada), Proceedings 63rd Canadian Geotechnical Conference and the 6th Canadian Permafrost, Calgary, Canada, 476–
483, https://doi.org/10.13140/2.1.2467.2961, 2010.

Steven, B., Briggs, G., McKay, C. P., Pollard, W. H., Greer, C. W. and Whyte, L. G.: Characterization of the microbial diversity in a permafrost sample from the Canadian high Arctic using culture-dependent and culture-independent methods, FEMS Microbiol. Ecol., 59, 513–523, https://doi.org/10.1111/j.1574-6941.2006.00247.x, 2007.

Steven, B., Pollard, W. H., Greer, C. W. and Whyte, L. G.: Microbial diversity and activity through a permafrost/ground ice core profile from the Canadian
high Arctic, Environ. Microbiol., 10, 3388–3403, https://doi.org/10.1111/j.1462-2920.2008.01746.x, 2008.

Strauss, J., Laboor, S., Schirrmeister, L., Grosse, G., Fortier, D., Hugelius, G., Knoblauch, C., Romanovsky, V. E., Schädel, C., Schneider von Deimling, T., Schuur, E. A. G., Shmelev, D., Ulrich, M. and Veremeeva, A.: Geochemical, lithological, and geochronological characteristics of sediment samples from Yedoma and thermokarst deposits in Siberia and Alaska 1998-2016, PANGAEA, https://doi.org/10.1594/PANGAEA.919064, 2020.

Subedi, R., Kokelj, S. V. and Gruber, S.: Ground ice, organic carbon and soluble cations in tundra permafrost soils and sediments near a Laurentide ice
divide in the Slave Geological Province, Northwest Territories, Canada, Cryosphere, 14, 4341–4364, https://doi.org/10.5194/tc-14-4341-2020, 2020.



Syromyatnikov I. I. and Kunitsky V. V.: The structure of lacustrine deposits in the cultural layer of the City of Yakutsk. Earth`s Cryosphere, 23, 13–21. https://doi.org/10.21782/EC2541-9994-2019-4(13-21), 2019.

Takano, S., Sugimoto, A., Tei, S., Liang, M., Shingubara, R., Morozumi, T. and Maximov, T. C.: Isotopic compositions of ground ice in near-surface permafrost in relation to vegetation and microtopography at the Taiga–Tundra boundary in the Indigirka River lowlands, northeastern Siberia, PLoS ONE, 755   14, e0223720, https://doi.org/10.1371/journal.pone.0223720, 2019.

Tanski, G., Couture, N., Lantuit, H., Eulenberg, A. and Fritz, M.: Eroding permafrost coasts release low amounts of dissolved organic carbon (DOC) from ground ice into the nearshore zone of the Arctic Ocean, Global Biogeochem. Cy., 30, 1054–1068, https://doi.org/10.1002/2015GB005337, 2016.

Tarnocai, C. and Bockheim, J.: Cryosolic soils of Canada: Genesis, distribution, and classification, Can. J. Soil Sci., 91, 749–762, https://doi.org/10.4141/cjss10020, 2011.

Trochim, E. D., Schnabel, W. E., Kanevskiy, M., Munk, J. and Shur, Y.: Geophysical and cryostratigraphic investigations for road design in northern Alaska, Cold Reg. Sci. Technol., 131, 24–38, https://doi.org/10.1016/j.coldregions.2016.08.004, 2016.

Vasil'chuk, A. C. and Vasil'chuk, Y. K.: Engineering-geological and geochemical conditions of polygonal landscapes in the area of the Tambey River mouth (Yamal Peninsula, northern Siberia), Inzhenernaya Geologiya, 4, 36–54, 2015.

Väisänen, M., Krab, E. J., Monteux, S., Teuber, L. M., Gavazov, K., Weedon, J. T., Keuper, F. and Dorrepaal, E.: Meshes in mesocosms control solute and 765   biota exchange in soils: a step towards disentangling (a)biotic impacts on the fate of thawing permafrost, Appl. Soil Ecol., 151, 103537, https://doi.org/10.1016/j.apsoil.2020.103537, 2020.

Walker, D. A., Buchhorn, M., Kanevskiy, M., Matyshak, G. V., Raynolds, M. K., Shur, Y. L. and Wirth, L. M.: Infrastructure-thermokarst-soil-vegetation interactions at Lake Colleen Site A, Prudhoe Bay, Alaska, Alaska Geobotany Center Data Report AGC 15-01, 92 pp., Institute of Arctic Biology, University of Alaska Fairbanks, Fairbanks, AK, 2015.

Wang, Q, Jin, H., Zhang, T., Cao, B., Peng, X., Wang, K., Xiao, X., Guo, H., Mu, C. and Li, L.: Hydro-thermal processes and thermal offsets of peat soils in the active layer in an alpine permafrost region, NE Qinghai-Tibet plateau, Global Planet. Change, 156, 1–12, https://doi.org/10.1016/j.gloplacha.2017.07.011, 2017.

Wang, W., Wu, T., Chen, Y., Li, R., Xie, C., Qiao, Y., Zhu, X., Hao, J. and Ni, J.: Spatial variations and controlling factors of ground ice isotopes in permafrost areas of the central Qinghai-Tibet Plateau, Sci. Total Environ., 688, 542–554, https://doi.org/10.1016/j.scitotenv.2019.06.196, 2019.

Wetterich, S., Kuzmina, S., Andreev, A. A., Kienast, F., Meyer, H., Schirrmeister, L., Kuznetsova, T. and Sierralta, M.: Palaeoenvironmental dynamics inferred from late Quaternary permafrost deposits on Kurungnakh Island, Lena Delta, northeast Siberia, Russia, Quaternary Sci. Rev., 27, 1523–1540, https://doi.org/10.1016/j.quascirev.2008.04.007, 2008.

Wetterich, S., Schirrmeister, L. and Kholodov, A. L.: The joint Russian-German expedition Beringia/Kolyma 2008 during the International Polar Year (IPY) 2007/2008, Berichte zur Polar-und Meeresforschung (Reports on Polar and Marine Research), 636, http://hdl.handle.net/10013/epic.38415, 2011.

Wetterich, S., Grosse, G., Schirrmeister, L., Andreev, A. A., Bobrov, A. A., Kienast, F., Bigelow, N. H. and Edwards, M. E.: Late Quaternary environmental and landscape dynamics revealed by a pingo sequence on the northern Seward Peninsula, Alaska, Quaternary Sci. Rev., 39, 26–44, https://doi.org/10.1016/j.quascirev.2012.01.027, 2012.

Wetterich, S., Tumskoy, V., Rudaya, N., Kuznetsov, V., Maksimov, F., Opel, T., Meyer, H., Andreev, A. A. and Schirrmeister, L.: Ice Complex permafrost of MIS5 age in the Dmitry Laptev Strait coastal region (East Siberian Arctic), Quaternary Sci. Rev., 147, 298–311, 785   https://doi.org/10.1016/j.quascirev.2015.11.016, 2016.

Wetterich, S., Davidson, T. A., Bobrov, A., Opel, T., Windrisch, T., Johansen, K. L., González-Bergonzoni, I, Mosbech, A. and Jeppesen, E.: Stable isotope signatures of Holocene syngenetic permafrost trace seabird presence in the Thule District (NW Greenland), Biogeosciences, 16, 4261–4275, https://doi.org/10.5194/bg-16-4261-2019, 2019

Windrisch, T., Grosse, G., Ulrich, M., Schirrmeister, L., Fedorov, A. N., Konstantinov, P., Fuchs, M. and Strauss, J.: Organic material, sediment and ice 790   characteristics of two permafrost cores from Yukechi Alas, Central Yakutia, PANGAEA, https://doi.org/10.1594/PANGAEA.898754, 2019.

Wolfe, S. A., Burgess, M. M., Douma, M., Hyde, C. and Robinson, S.: Geological and geophysical investigations of ground ice in glaciofluvial deposits, Slave Province, District of Mackenzie, Northwest Territories, Geological Survey of Canada, Open File 3442, 110 pages, https://doi.org/10.4095/208917, 1997.



Wolfe, S. A.: Massive ice associated with glaciolacustrine delta sediments, Slave Geological Province, N.W.T. Canada, 7th International Permafrost Conference, Yellowknife, Canada, Collection Nordicana, 55, 1133–1139, 1998.

Wolfe, S. A., Duchesne, C., Gaanderse, A., Houben, A. J., D'Onofrio, R. E., Kokelj, S. V. and Stevens, C. W.: Report on 2010-2011 Permafrost Investigations in the Yellowknife Area, Northwest Territories, GSC Open file 6983, https://doi.org/10.4095/289596, 2011.

Wolfe, S. A., Stevens, C. W., Gaanderse, A. J. and Oldenborger, G. A.: Lithalsa distribution, morphology and landscape associations in the Great Slave Lowland, Northwest Territories, Canada. Geomorphology, 204, 302–313, https://doi.org/10.1016/j.geomorph.2013.08.014, 2014.

Wu, T., Wang, Q., Watanabe, M., Chen, J. and Battogtokh, D.: Mapping vertical profile of discontinuous permafrost with penetrating radar at Nalaikh depression, Mongolia, Environmental Geol., 56, 1577–1583, https://doi.org/10.1007/s00254-008-1255-7, 2009.

Yu, F., Qi, J., Yao, X. and Liu, Y.: In-situ monitoring of settlement at different layers under embankments in permafrost regions on the Qinghai–Tibet Plateau, Eng. Geol., 160, 44–53, https://doi.org/10.1016/j.enggeo.2013.04.002, 2013.

Zheng, M., Yang, Z., Yang, S. and Still, B.: Modeling and mitigation of excessive dynamic responses of wind turbines founded in warm permafrost, Eng.
Struct., 148, 36–46, https://doi.org/10.1016/j.engstruct.2017.06.037, 2017.



**Author contributions.** OK and JH developed the original idea. OK led the compilation of observational data with

contributions from MZK and JH, and geospatial data processing with JA. OK performed the statistical analyses with

contributions from JH, JA and ML. OK wrote the manuscript with contributions from all the authors.

**Competing interests.** The authors declare that they have no conflicts of interest.

**Financial support.** OK and JH acknowledge funding from the Academy of Finland (AoF; Grant 315519). JA acknowledges

the AoF Flagship funding (grant no. 337552). MZK acknowledges funding from the National Science Foundation (grants

OPP-1820883, OPP-1806213, NNA-1928237, NNA-2126965, and NNA-2022590).





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
