# Peer review of "High-resolution predictions of ground ice content for the Northern Hemisphere permafrost region"

_Earth System Science Data, 2022_

## Referee Comment (RC1)

**Overview**

The submitted article presents predictions of volumetric ice content (indicated as the sum of pore and segregated ice content) in the top 5 m of permafrost across the entire circum-Arctic, at a higher resolution (~1 km) than previous products covering large geographic extents. These predictions are based on statistical modelling using datasets on grain size, modern climate indices (thawing degree days, freezing degree days, snowfall), modern vegetation conditions based on the Normalized Difference Vegetation Index (NDVI), a computed topographic wetness index, a model of the probability of bedrock in the upper 200 cm, and the % coverage of water bodies. The authors indicate that the data may be useful for environmental impacts assessment applications (l. 24).

I commend the authors on the effort compiling a dataset of ice contents from the literature. The aim of the paper (improving spatial representation of ground ice conditions) is indeed an important topic worthy of continued efforts due to the implications of ice-rich permafrost thaw on infrastructure, landscape change, hydrology, and the global carbon budget. However, the methodology used in this article is inappropriate for predictions of ice content in the upper 5 m of permafrost. Consequently, the predictions depart from current knowledge of ground ice conditions in many areas, and past generalized modelling. Furthermore, the scale of the modelling makes it unsuitable for use in environmental impact assessments, and in such applications, appreciation of all types of ground ice is required. Therefore, I cannot recommend this paper for publication, and strongly caution against the use of this product for environmental impact assessments.

**Ground ice modelling approach**

The volumetric ice content in the upper 5 m of permafrost includes, among other forms, pore ice and segregated ice (the two considered in this paper). Pore ice volume is controlled by the porosity of the material and degree of saturation. Segregated (and pore) ice accumulation over this depth range, as the authors point out, may occur over millennia (l. 393). Segregated ice is of particular consequence to infrastructure risk assessments as it commonly occurs in excess of the soil's natural pore space near the top of permafrost within fine-grained sediments. Thaw of ice-rich permafrost results in differential consolidation and subsidence of the ground surface and may contribute to slope instability, posing hazards to infrastructure. The abundance of segregated ice in upper permafrost is controlled by variables including soil moisture conditions (during permafrost aggradation, and subsequently), soil texture (frost susceptibility), the mode of permafrost aggradation, and variations in the permafrost table over long time periods (e.g., French and Shur, 2010; O'Neill et al. 2012; Gilbert et al. 2018; Cheng 1983, and references therein).

**Predictor variable selection**

The modelling in this article includes many variables with little or no clearly demonstrated relevance to ground ice formation over millennia, and their inclusion in the model is not justified in text with reference to previous work. For example, the process link between segregated ice formation and modern climatic conditions (FDD, TDD, snowfall) and modern vegetation conditions represented by NDVI are not defined. These predictor variables do not account for the mode of permafrost aggradation

(epigenetic, syngenetic), variation in the permafrost table over time due to climatic shifts or disturbance, or geomorphic processes that influence permafrost and ground ice aggradation. Therefore, from a very basic process level, the model is unlikely to yield reliable predictions of segregated ground ice content.

The process relation between water body % cover and ground ice is also unclear. The authors indicate "The effects of water bodies, whether hydrothermal or due to the spatial association between thermokarst lakes and ice-rich environments, were accounted for…" p.8. Hydrothermal effects (i.e., the presence of taliks[?]) would presumably lower ground ice content in pixels that have many water bodies, but the association made between thermokarst lakes and ice-rich environments suggests the opposite (that there would be higher ground ice content in areas with many thermokarst lakes). This is contradictory. Furthermore, not all waterbodies, and in fact many or most in some regions (e.g., the Canadian Shield), are not formed by the thaw of ice-rich permafrost, and thus are not thermokarst lakes, so the link between ice-rich environments and lakes is poorly founded. The methodology makes no distinction between lakes of different origin or in settings with different surficial material properties.

Snowfall is redistributed by wind over much of the Arctic, and snow cover accumulation is largely controlled by topographic and vegetation conditions, rather than snowfall itself. No explanation is given as to the relevance of snowfall to VIC.

In relation to predictor variable selection, I note that Karjalainen et al 2019 and 2020 are cited in text to justify the modelling approach in circumarctic contexts (l. 233), but neither are included in the reference list. Karjalainen et al 2020 also present statistical models in which predictor variables in the model (of high "relative importance", Fig 4a) bear little relevance to the formation of the landforms of interest. For example, rainfall was the most "important" variable in the pingo modelling. Closed system pingo formation – the common mode in the western Canadian Arctic, and represented by many observations in the training dataset – depends on 1) air temperatures that promote permafrost aggradation following subaerial exposure of sediments, 2) the drainage of (typically) lakes underlain by closed taliks, and 3) sandy substrates within the taliks, which subsequently refreeze (e.g., Mackay, 1987). These conditions bear no relation to modern rainfall; the substrate into which freezing occurs during pingo formation is saturated due to the former presence of the lake in a topographic low, and downward freezing and permafrost formation proceeds following subaerial exposure of the lake bottom. Therefore, the "relative importance" of rainfall is purely correlative due to the range of rainfall conditions where pingos are observed. Consequently, based on climate change scenarios, the authors present predictions for "new environmental space" for pingos over vast areas of the Canadian Arctic Archipelago where geomorphic conditions would preclude their formation (Figure 2, Karjalainen et al. 2020). The purpose of raising this example from a prior publication is to highlight the deficiencies in this type of statistical modelling for applications in which the predictor variables have little or no physical bearing on the periglacial processes being considered. I note that relative importance of predictor variables is not reported in the submitted paper.

**Scale**

As the authors acknowledge on p. 14: "Ground ice content can display high spatial variability over small horizontal distances". Dr. L.U. Arenson correctly points out in the comments on the ResearchGate preprint webpage (https://www.researchgate.net/publication/363498625_High-

), that the scale at which the modelling is presented (~1 km) is unable to represent this complexity, making it unsuitable for risk assessments for northern infrastructure, which require observations at the site scale. Quoting Dr. Arenson: "The authors use the argument that current information on ground ice is insufficient for hazard assessments of Arctic Infrastructure (line 12) to justify their product, implying indirectly that their map can now be used for exactly that purpose". This will certainly lead to misuse of the product.

**Input data:**

Ground ice at the top of permafrost is predicted for large regions of the southern Canadian prairies (west of Lake Winnipeg), where no permafrost exists**.** The authors acknowledge "According to Obu et al. (2019) the extent of isolated permafrost in their zonation may be overestimated in eastern Russia and central Canada, especially. Such areas may not be underlain by permafrost and therefore *associated ground ice predictions are not applicable, but also not in the focus of the produced data*." (p.15). This remark is curious, as the authors suggest that accuracy over large regions is not a focus of the output, despite the assertion in the abstract that outputs allow "consideration of ground ice content in various geomorphological, ecological, and environmental impact assessment applications". Environmental impact assessments require accurate characterization of ground ice. If the ground ice predictions are not applicable over large areas, why are they included? Broad-scale predictions without a focus on accuracy, or the accuracy of datasets used to produce the predictions, are not suited to environmental impacts applications.

The accuracy of other input datasets is also questionable. For example, the Rhorizon layer was produced using a similar statistical modelling approach (Shangguan et al. 2017). The training datasets based on soil profiles and drilling logs is shown in the image below (Figure 2). There is a distinct lack of training data over large regions of the permafrost domain, namely the Canadian Arctic Archipelago, the Canadian Shield portion of the mainland, and vast areas of Siberia. Shangguan et al. (2017) provide calibration and validation metrics for 4 Canadian provinces, but none for the 3 territories, where most permafrost is found. These are the areas where training data is most sparse, or, as is the case over much of the Canadian Arctic Archipelago, almost entirely lacking. No model performance statistics are provided for Asia, and the authors indicate that training data sparsity is an issue for accuracy there (Sect. 4.7). Logically, this suggests that RHorizon accuracy is an issue over most permafrost regions. Therefore, in addition to the TTOP permafrost layer being inaccurate in southern permafrost zones, the accuracy of the RHorizon predictor may be poor across circumpolar regions. No discussion of the implications of using such models as predictor variables is provided, other than one sentence suggesting that "More spatially resolved climate and soil predictors could offer a straightforward but not easily attainable means to improve prediction accuracy of the models." (l. 438).

[Figure]

Figure 2 from Shangguan et al. (2017): Global distribution of depth to bedrock observations. (a) Red colors indicate soil profiles, (b) blue colors boreholes.

**Training data**

The authors state that "On average, 251 VIC observations were used in the 100 calibration runs (Fig. A2) and 44 in the evaluation" p.8. The 251 observations used in calibration runs amounts to one observation per 55,378 sq km of permafrost area (based on $13.9 \times 10^6$ km$^2$ from Obu et al. 2019). Given the high variation in ground ice content over different spatial scales, the training dataset seems insufficient for statistical modelling, particularly considering the clustering of observations. As an applied example, let us assume that the southern prairies west of Lake Winnipeg did actually have permafrost. Predicted VICs in this region are 10-30% (Figure D5). Among 34 observation sites with VIC <30%, 5 are from bedrock in the Alps (VIC = 0%), 9 are from the Tibetan Plateau (arid steppe), and 6 are from Mongolia (also arid). Substrate conditions are drastically different at these sites than in the Canadian prairies. Five sites from Canada have VIC <30%, but four of these are on the Canadian shield, where surficial materials differ significantly from the prairies, and are generally much thinner, and one is from an Arctic setting on Banks Island. The sites in Alaska and Russia are similarly in Arctic settings. Therefore, there are no VIC training data that represent substrate conditions over this large region for the ice contents predicted there (10-30%). As such, there is little evidence to suggest that the training data is sufficient to reasonably represent ground ice conditions worldwide or the range in conditions that control its distribution. For comparison, the Shangguan et al (2017) modelling includes millions of observations, vs. hundreds in this modelling, and still has large regions that are drastically underrepresented and cause accuracy issues acknowledged by those authors.

Peatlands. Many of the observations in the training dataset that have the highest ice contents (e.g., >80% VIC) are from peatlands. For example, observations from Smith et al. 2012 were from "Sphagnum-dominated raised bogs and peat plateaus". The sampling did not include underlying mineral material. Therefore, these high VICs are due to the high porosity of peat material, though they are used to train a model that is predicting pore and segregated ice, but which does not include a predictor variable that considers peat coverage or depth. Though the 'fine' predictor may be correlated with peatland presence, this does not address the fact that high VICs can be caused by highly porous material (peat) or a combination of (relatively lower) porosity in fine-grained mineral material and significant fractions of excess ice. The modelling cannot make this important distinction, and the significant training data from peatlands will affect outputs for areas where other predictor variable values are similar, but where peatlands may not exist (e.g., over the Canadian Shield, discussed below).

The data from Illisarvik appear to be from the 26 drained lake basin sites. Ground ice content is not representative of the surrounding tundra that comprises most of the terrestrial landscape, as it has only accumulated in the decades since lake drainage, not over millennia as in the tundra. From what I can tell, this is the only data point for Richards Island.

In summary, the training data do not adequately capture conditions over the model domain, and the modelling cannot differentiate between high VICs caused by peat presence vs. from segregated ice formation in (dominantly) fine-grained mineral soils.

**Comparison with previous models**

The authors indicated "The VIC predictions show a general match with the prominent high- and low-ice content areas in the reference maps but with a greater spatial variability (Fig. 6)." p. 12.

-The Jorgenson map is of excess ice from all ice types, making the comparison largely irrelevant.

-There are large areas where the ice predictions differ significantly from the segregated ice map by O'Neill et al. 2019. For example, in Quebec:

[Figure]

[Figure]

-The above is on the Canadian shield, which includes dominantly thin, coarse-grained glacial deposits. Therefore, the porosity is relatively low and the sediments are not typically frost-susceptible. The authors do acknowledge the undertainty in this area on l. 410 " Moreover, some areas, such as parts of the central **Ungava Peninsula**[***] in Canada (Fig. 4a), based on the used data have very low fine-grained sediment (sum of Clay and Silt fractions, Table 2) contents (<300 g kg-1) that fall outside the range covered by the modelling data. As a result, associated predictions have relatively high

uncertainty". This acknowledges that the data used in the modelling does not represent conditions on the Canadian Shield, which covers a total area of 8,000,000 km$^2$, with a significant fraction falling within permafrost regions. Therefore, in addition to acknowledged accuracy/uncertainty in sporadic and isolated permafrost zones, accuracy is also likely poor over the vast Canadian Shield.

***The area marked as (9) Ungava Peninsula on e.g., Figure D5 is not Ungava Peninsula. Ungava Peninsula is marked below (https://en.wikipedia.org/wiki/Ungava_Peninsula):

[Figure]

[Figure]

-Predictions west of Hudson Bay (north of Hudson Bay Lowlands) also indicate higher values over large areas of the Canadian Shield where previous modelling typically indicates low excess ice abundance in (mainly) coarse grained tills, other than in areas near the coast that were previously submerged and include fine-grained marine sediments.

[Figure]

-Banks and Victoria Island. On Banks Island, for example, ice content is generally greater on the eastern side of the island on the O'Neill et al (2019) mapping, and lower on the western coast, however, the presented output shows a generally opposing pattern. There are also notable differences on Victoria Island.

[Figure]

-Broad patterns from the IPA map of relatively high ice abundance on Brodeur Peninsula (NW Baffin Island), also represented in the modelling by O'Neill et al. 2019, and western Southhampton island are not in general agreement with predictions in this study.

In my opinion, indicating a "general match" between patterns glosses over significant deviations in many areas, and the differences in what the products represent.

**Summary**

In summary, an effective statistical model should include variables that are related, through established physical processes, to the phenomenon being modelled. For many of the variables in this exercise, this is not the case. The training data should also be sufficient to capture variation in conditions across the modelled domain. This has not been demonstrated. Data layers known to be inaccurate are used as inputs, and overall cause implausible outputs over large geographic areas. The authors indicate: "The predictive performances of the models (R2) show a clear drop when calibrated models are used to predict VIC at sites in the spatially independent evaluation datasets…" P. 290. This indicates a model with limited predictive utility. The authors also assert on l. 400 "Despite the discussed uncertainties, we argue that attributed to the detailed observational data and careful harmonization procedures with uncertainty estimations, the produced datasets provide a reasonably accurate account of ground ice

content for modelling and validation purposes at regional and circumarctic scales". This seems highly optimistic considering the evidence to the contrary.

Based on the material presented, the modelling has not demonstrated an advance in knowledge on the distribution of ground ice, or convincingly demonstrated general agreement with previous broad-scale modelling. The modelling approach and selected predictor variables indicates a lack of process knowledge, and against evaluation datasets, performs poorly. The relative importance values of predictor variables in the model are not indicated. The outputs are demonstrably inaccurate in many areas. Based on the material I've mentioned above, and stated by the authors in the paper, poor accuracy can be expected in 1) sporadic/discontinuous permafrost zones, 2) the Canadian Shield, and 3) Yedoma regions. In some areas, the output may be reasonably accurate but for the wrong reasons, due to the consideration of pore and segregated ice together. These predictions should not be used in environmental impact assessments, nor should other outputs from broad-scale models.

Brendan O'Neill

**Other:**

Many cited works are not in the reference list. E.g., O'Neill et al. 2019, Karjalainen et al. 2019 & 2020,

l. 62. This implies frost susceptibility is related to differential evaporation/sublimation based on grain size, which is inaccurate.

Line 258: 2500 cm?

Line 207. Zero explanation of why modern FDD and TDD are relevant to pore and segregated ground ice abundance formed over millennia.

NDVI: There is no indication how this is expected to relate to ground ice content.

Line 340. The model does not actually reflect thick sediments since you are only considering the soil texture from 100-200 cm.

Supplementary. The column "country" includes entries that are not countries (The Alps, Alaska).

Mean Error is a relatively useless measure of accuracy as positive and negative errors cancel each other out.

**References**

Cheng, Guodong. 1983. The mechanism of repeated-segregation for the formation of thick layered ground ice. Cold Regions Science and Technology, 8(1), pp.57-66.

French, H. and Shur, Y., 2010. The principles of cryostratigraphy. Earth-Science Reviews, 101(3-4), pp.190-206.

Gilbert, G.L., O'Neill, H.B., Nemec, W., Thiel, C., Christiansen, H.H. and Buylaert, J.P., 2018. Late Quaternary sedimentation and permafrost development in a Svalbard fjord-valley, Norwegian high Arctic. Sedimentology, 65(7), pp.2531-2558.

Karjalainen, O., Luoto, M., Aalto, J., Etzelmüller, B., Grosse, G., Jones, B.M., Lilleøren, K.S. and Hjort, J., 2020. High potential for loss of permafrost landforms in a changing climate. Environmental Research Letters, 15(10), p.104065.

Mackay, J.R., 1987. Some mechanical aspects of pingo growth and failure, western Arctic coast, Canada. Canadian Journal of Earth Sciences, 24(6), pp.1108-1119.

Obu, J., Westermann, S., Bartsch, A., Berdnikov, N., Christiansen, H.H., Dashtseren, A., Delaloye, R., Elberling, B., Etzelmüller, B., Kholodov, A. and Khomutov, A., 2019. Northern Hemisphere permafrost map based on TTOP modelling for 2000–2016 at 1 km2 scale. Earth-Science Reviews, 193, pp.299-316.

O'Neill, H.B. and Burn, C.R., 2012. Physical and temporal factors controlling the development of near-surface ground ice at Illisarvik, western Arctic coast, Canada. Canadian Journal of Earth Sciences, 49(9), pp.1096-1110.

Shangguan, W., Hengl, T., Mendes de Jesus, J., Yuan, H. and Dai, Y., 2017. Mapping the global depth to bedrock for land surface modeling. Journal of Advances in Modeling Earth Systems, 9(1), pp.65-88.

Smith, L.C., Beilman, D.W., Kremenetski, K.V., Sheng, Y., MacDonald, G.M., Lammers, R.B., Shiklomanov, A.I. and Lapshina, E.D., 2012. Influence of permafrost on water storage in West Siberian peatlands revealed from a new database of soil properties. Permafrost and Periglacial Processes, 23(1), pp.69-79.